# Implementation of safety management systems (SMS) in healthcare: a systematic review and international comparison

Siân de Bell , Zhivko Zhelev , Alison Bethel, Maria Clarke, Rob Anderson, Jo Thompson Coon

SdB and ZZ are joint first authors.

Exeter HSDR Evidence Synthesis Centre, Department of Health & Community Sciences, University of Exeter Medical School, University of Exeter, Exeter, UK

**Correspondence to**
Dr Siân de Bell;
s.c.de-bell@exeter.ac.uk

## ABSTRACT

**Objectives**  Many industries where safety is a priority (eg, aviation) use safety management systems (SMSs), but evidence on their use in healthcare is needed to determine whether they could support patient safety improvement. We investigated the application of national-level SMSs to patient safety in terms of effectiveness, implementation and experience.

**Design**  Systematic review using a case study approach.

**Data sources**  We identified patient safety organisations in each country and searched their websites. We also searched MEDLINE (in December 2023) and Embase (via Ovid), CINAHL (EBSCO) and Web of Science (in February 2024).

**Eligibility criteria**  Any evidence from five high-income countries that have publicly funded healthcare systems with universal coverage: Australia, Canada, Ireland, New Zealand and the Netherlands. We included publications on the effectiveness, implementation or experience of the patient safety approach in each country.

**Data extraction and synthesis**  Included evidence was summarised and mapped onto an initial analytical framework based on analysis of SMSs in high-risk industries, enabling cross-country comparisons. Drafts were shared with experts from patient safety organisations in each country for comment. Formal quality appraisal was not possible as most evidence was non-evaluative.

**Results**  53 publications were included, from Australia (5), Canada (7), Ireland (8), New Zealand (9) and the Netherlands (24). Only the Netherlands implemented a patient safety programme explicitly based on an SMS approach. Some aspects of patient safety in hospitals improved as a result, but there was significant variation in both the implementation of the programme and attributed outcomes.

In the other four countries, the main components of an SMS were identified to some extent, along with evidence that their patient safety approaches had been influenced by concepts from high-risk industries and 'safety science' more widely.

**Conclusions**  Emerging ideas from high-risk industries (beyond SMSs) and broader safety science have influenced all countries, reflecting increasing awareness of the need for initiatives to be context-specific to be successful. However, their implementation and impact need further evaluation.

## STRENGTH AND LIMITATIONS

⇒ This systematic review developed a framework defining the key components of a safety management system, from their use to improve safety in high-risk industries (eg, aviation), and applied this to analyse national patient safety approaches in healthcare.

⇒ We included a broad evidence base, searching the websites of patient safety organisations alongside searching for peer-reviewed articles and consulting safety experts in included health systems to verify our understanding of their approach to safety management in healthcare.

⇒ We focused on high-income countries with healthcare systems comparable to the UK. However, we were only able to include those where evidence (eg, policy documents) was available in English.

⇒ Our analysis was based mainly on non-evaluative documents, so we were not able to conduct a quality appraisal or make firm conclusions about the impacts of adopting a safety management system approach in healthcare.

**PROSPERO registration number**  CRD42023487512.

## BACKGROUND

All healthcare systems work in various ways to improve patient safety, from creating policies at national and organisational levels to implementing interventions as a result. The challenges of ensuring safe healthcare are recognised by the WHO, which has developed the Global Patient Safety Action Plan 2021–2030 to provide countries with a framework to improve their national patient safety approaches.[1] However, patient safety improvement is not an easy process; for example, a national review of patient safety in England in 2024 found that, compared with the previous review in 2022, performance had worsened in a number of areas (eg, maternity care).[2] Globally, patients continue to be harmed by failures or errors, with examples

of safety incidents including mistakes associated with the use of medication, such as in prescribing or dispensing, and wrong-site surgery.[3] In high-income countries, 1 in 10 patients are thought to be subject to an adverse event while receiving hospital care each year,[1] while in England, it is estimated that 237 million medical errors occur annually, contributing to more than 1700 deaths.[4]

Safety is also a significant priority in other industries, such as aviation, nuclear energy and oil and gas, where errors could have serious consequences for people or the environment.[3] Many of these 'safety-critical' industries, and companies within them, have improved safety by taking a systems approach, considering the many interacting organisation and process-level factors (eg, culture, technology and team relationships) that may result in errors or adverse events.[1] This has led to the development of international or national policies and standards for safety, with organisations operationalising these in an organised approach to managing safety known as a safety management system (SMS).[5] There is no single definition for an SMS, but they are generally considered to have four key components.

► Leadership commitment and safety policy—an expressed leadership commitment to safety with documentation of responsibilities and processes for safety within an organisation.[3 6]
► Safety risk management—the identification of hazards and risks and assessment of how to mitigate these.[3]
► Safety assurance—monitoring and measuring safety within the organisation and ensuring continuous improvement[3].
► Safety promotion and culture—training, education and communication of safety to staff at all levels.[7]

The underlying theory behind SMSs is that by focusing on performance—monitoring and achieving desired outcomes—rather than solely on compliance, they facilitate improvement.[3] In 2012, a systematic review of SMSs in three safety-critical industries—aviation, marine and rail—concluded that the approach led to improvements in safety (eg, reduced accident rates).[8] This suggests that they have the potential to improve safety in other sectors where safety is paramount, such as healthcare. The WHO Global Patient Safety Action Plan 2021–2030 recognises 'the strong parallels (of healthcare) with the experience of other high-risk industries, thus creating opportunities for transfer learning' (p. 2).[1]

The basic components of an SMS (as described above, with further detail given in table 1) are quite generic and transferable.[9] It is not possible to provide a more detailed description of a single SMS, as in order to be effective, the specifics need to be adapted to the context in which they are being used, as can be seen from SMSs in other high-risk industries.[10] For example, the aviation, nuclear and oil and gas industries all have international guidelines regarding safety management.[6 7 11] In aviation, these are used by countries to develop state safety programmes with detailed requirements for safety management (ie, SMSs) by individual organisations. Nuclear safety is considered a national policy issue, with states responsible for the safety of their own nuclear facilities. The oil and gas industry differs from both in that it identifies and manages safety risks within the context of achieving performance goals and stakeholder benefits.[11] In all these industries, individual organisations implement policy and guidance according to their own ways of working.

While healthcare systems globally are working to improve patient safety, countries have different starting points in terms of the organisation and management of their healthcare and different pre-existing system components (eg, approaches to monitoring patient safety incidents). Most healthcare systems have some components of an SMS, such as policies for improving patient safety,[12] with some countries, notably the Netherlands,[13] explicitly developing and promoting an SMS approach. Their experiences offer opportunities for learning and determining whether the principles of SMSs are effective in improving safety in healthcare and, if so, how they can be adapted from industries in which they are used and implemented within a healthcare system. In this review, we focus on the high-level influence or coordination of the implementation of an SMS approach, rather than the operation of organisation-level SMSs.

## Objectives
This systematic review investigated the application of SMSs to patient safety in healthcare to consider whether and how they could be implemented more widely, asking:
► How are the components of an SMS reflected in the healthcare policy documents of the included countries?
► What research or other relevant evidence is available regarding the effectiveness, implementation or experience of SMSs within healthcare?
► What does evidence from beyond the UK tell us about the effectiveness, implementation or experience of SMSs within healthcare?

This review was commissioned by the National Institute for Health and Care Research Health Service Delivery Research (HSDR) programme on behalf of the National Health Service (NHS) England Patient Safety Team and Department of Health and Social Care to inform the further development of NHS England's patient safety policy and practice. The UK was not included in the review as (1) the NHS has not implemented an SMS and (2) the intention was to learn from the experiences of other countries, with different starting points in terms of the organisation and management of healthcare and different pre-existing (safety) system components, regarding their patient safety approaches.

## METHODS
This systematic review was registered on PROSPERO (CRD42023487512) and can be found in online supplemental file 1. A summary of the methods is provided below; further details can be found in the full report.[14]

**Table 1** Framework of the components of an SMS

| Key component | Subcategory | Definition |
|---|---|---|
| Leadership commitment and safety policy | Management commitment and accountability | Individuals in the organisation who are accountable for safety and their exact role and responsibilities should be clearly identified.[3] These individuals should ensure that the people they are managing have the capabilities to complete their required activities safely and provide support for them to do this (eg, resources).[6] |
| | High-level (policy) documents | The safety policy should outline the aims and objectives of an organisation in terms of safety.[3] |
| | External and internal boundaries of SMS | External boundaries might include requirements for contractors to have an SMS in place or be considered within the SMS of the organisation. Internally, the relation of the SMS to other management systems (eg, quality) needs to be defined. |
| | Coordination of the emergency response plan | Planning for immediate activities in response to an unplanned event, for example, an accident.[7] |
| Safety risk management | Hazard identification processes | There should be formal processes, of which there are likely to be more than one (eg, audits and incident reporting), in place to identify hazards.[7] |
| | Risk assessment and mitigation processes | This will identify the acceptability of the risk and appropriate measures to control it.[6] |
| Safety assurance | Safety performance monitoring and measurement | This involves the development of measures, both reactive and proactive, to routinely monitor safety performance within the organisation. It may also involve periodic audits and reviews of performance (including compliance), whether internally or externally (eg, by regulators). |
| | Internal safety investigations | Systems are needed to allow the reporting of safety incidents and processes to investigate their causes. |
| | Continuous improvement | Organisations should have arrangements to ensure that actions are taken if issues are identified as a result of monitoring and investigations, including ensuring that these are completed within appropriate timescales. |
| Safety promotion and culture | Training and education | Training may be specific to an individual's role, or to gain a wider understanding of safety issues, but is essential to ensure competence.[6] It should be identified, documented and reviewed and revised regularly. |
| | Safety communication | This may be formal (eg, safety meetings), or informal (eg, mention of issues in team briefings), but needs to be open and allow individuals to understand and accept why standards of safety are required and provide feedback.[6] |
| | Presence of safety culture | Also referred to as a just culture, allowing open communication and reporting of incidents or hazards. |

SMS, safety management systems.

### Inclusion and exclusion criteria

Our main criterion for inclusion was the national implementation of an SMS approach in the healthcare system; our scoping searches provided indicative evidence of the adoption of an SMS approach, or some of its key components, to improve patient safety in a few countries. We also decided to restrict the review to countries that were comparable to the UK and each other in that they are high-income countries[15] with national healthcare systems offering universal coverage (via general taxation or social insurance).[16 17] Consequently, evidence from the following countries was eligible for inclusion in the review: Ireland, Australia, New Zealand, Canada and the Netherlands. Additionally, the majority of evidence from these countries was available in English, and we had limited ability to translate non-English documents.

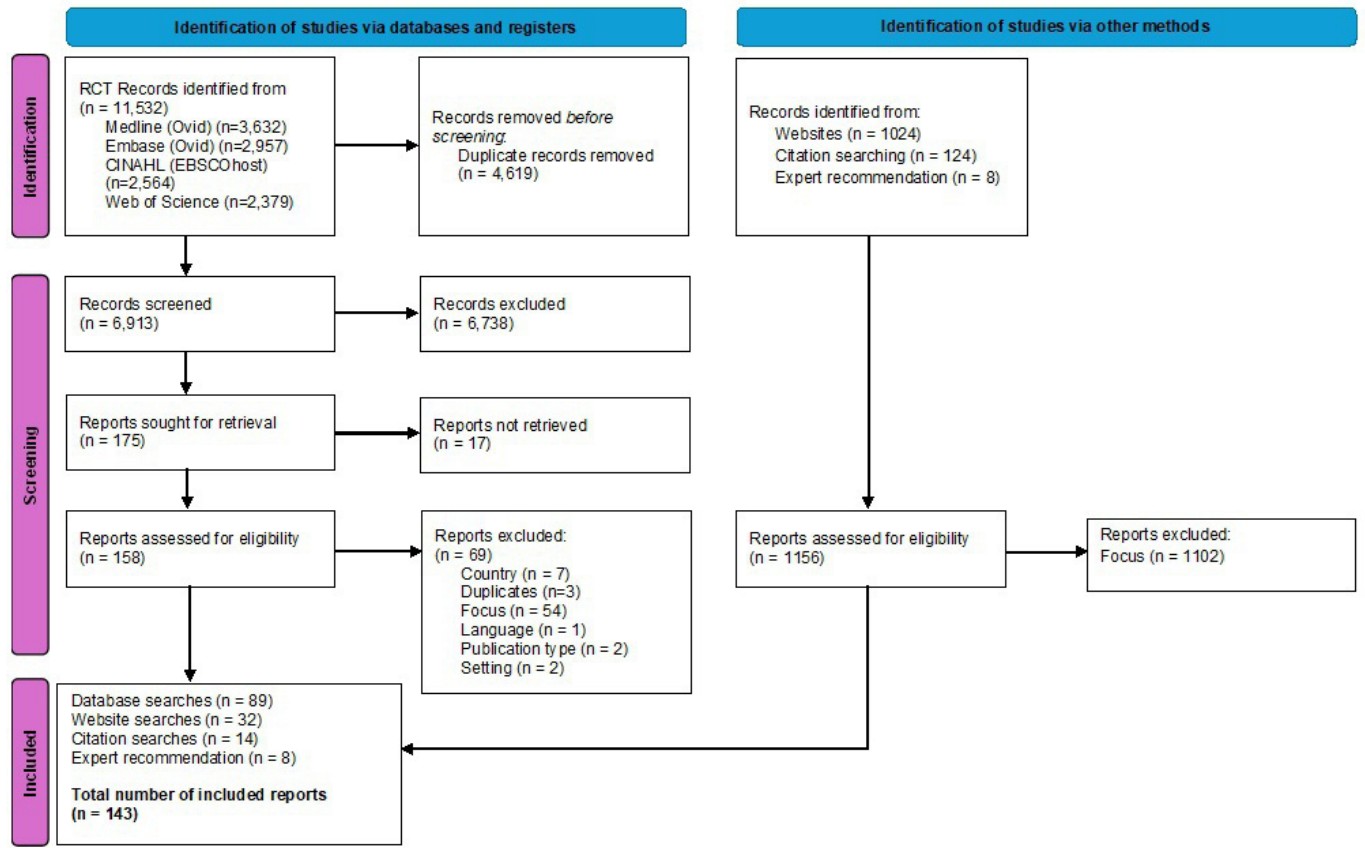

**Figure 1** Preferred Reporting Items for Systematic Reviews and Meta-Analyses (PRISMA) diagram of the screening process.

As this is an emerging, policy-relevant topic, we included a broad range of types of evidence that should contain evidence relevant to our review questions[18]:

▶ Policy documents, including those related to the implementation of an SMS or patient safety approach.[19]
▶ Published or unpublished research/evaluations (quantitative, qualitative or mixed methods).
▶ Other evidence (eg, patient and staff experience from organisations' websites).

Evidence relating to any service user population and any healthcare setting was included in the review; study participants could include healthcare professionals, service users or carers. In terms of focus, we included evidence on any policies, initiatives or programmes related to the key components of an SMS (as defined above and in table 1) or patient safety approach. Studies were excluded where their focus was not linked to a component of an SMS or was not patient safety (eg, occupational health and safety). See online supplemental file 1 for more information.

### Search strategy and study selection

MEDLINE (from 1946) was searched in December 2023 and Embase (from 1974) (via Ovid), CINAHL Ultimate (from 1937) (EBSCO) and Web of Science (WoS): Science Citation Index (from 1990), Social Science

**Table 2** Number of evidence sources per category per country

| Country | 'Rich' documents (n=53) | | | 'Thin' documents (n=90) |
|---|---|---|---|---|
| | Policy | Research | Other* | |
| Netherlands | 2 | 20 | 2 | 22 |
| Australia | 3 | 2 | 0 | 37† |
| Canada | 5 | 2 | 0 | 27† |
| Ireland | 7 | 1 | 0 | 2 |
| New Zealand | 9 | 0 | 0 | 6† |

*Such as opinion papers.
†Some included evidence had information from multiple countries (Australia and Canada (n=2) and Australia and New Zealand (n=2)) so has been included in multiple categories.

de Bell S, *et al. BMJ Open* 2026;**16**:e107772. doi:10.1136/bmjopen-2025-107772

Citation Index (from 1900), Arts and Humanities Index (from 1975), Conference Proceedings, Science, Social Science and Humanities (from 1990) and Emerging Sources Citation Index (from 2015) in February 2024 (see search strategies in online supplemental file 1). In March 2024, forward citation searching was undertaken in Web of Science using eight papers (from the Netherlands, as the only country explicitly using an SMS approach). The reference lists of all publications that met our inclusion criteria were checked for relevant publications.

Key policy documents and evidence that might not be captured through database searches (eg, programme reports) were identified by searching the websites of, and contacting, national patient safety organisations from each of the included countries (online supplemental file 1). To understand the context within each country and identify further sources of evidence, we also held online meetings with patient safety/health policy experts from:

► Australia—Australian Commission on Safety and Quality in Healthcare; Department of Health and Aged Care (federal government).
► Canada—Healthcare Excellence Canada.
► Ireland—Health Information and Quality Authority and Royal College of Surgeons Ireland.
► Netherlands—Netherlands Institute for Health Services Research.
► New Zealand—Health Quality and Safety Commission.

All results from the database searching were independently screened by two reviewers (SDB and ZZ). The full text of papers which were judged to meet the inclusion criteria was obtained, along with policy and other documents identified by web-searching. Two reviewers (SDB and ZZ) assessed each full text independently for inclusion. At both stages, disagreements were settled through discussion.

In December 2025, we checked that patient safety policy documents from the included countries remained the same and conducted further forward citation chasing on included studies.

### Data extraction and synthesis

While creating the review protocol and searching for studies, we developed an analytical framework of the components of an SMS by reviewing policy documents and review articles from high-risk industries (focusing on aviation,[7] oil and gas[11] and nuclear energy,[6] see table 1). Data extraction and thematic analysis of evidence from the healthcare systems of each included country were conducted by two reviewers (SDB and ZZ), who held regular discussions with the wider team and were guided by this framework.

Our protocol specified that we would only include countries which explicitly stated that they were using an SMS approach in their healthcare system, with our scoping searches suggesting that Australia, Canada, Ireland, the Netherlands and New Zealand all met this criterion. However, after screening studies for inclusion, we found that the Netherlands was the only country where this commitment to an SMS approach was explicit. Consultation with our main stakeholders, the NHS Patient Safety Team, indicated that understanding 'the principles underpinning SMS may help healthcare organisations to respond more effectively to patient safety risks' (p. 8).[3] We therefore decided to make a change to our protocol, continuing to include the other four countries in the review and extracting data on their approaches to patient safety and how these mapped to the components of an SMS.

An additional change was made to the protocol, given the quantity of studies retrieved by the database

**Table 3** Comparison of the scope of the SMS or safety approach in each country

| | Netherlands | Australia | Canada | Ireland | New Zealand |
|---|---|---|---|---|---|
| Geographical | National coverage. | National standards; state and territory governments are responsible for their own healthcare systems. | National framework; provincial and territorial governments are responsible for their own healthcare systems. | Standards and patient safety strategy are set at the national level but implementation is local to allow for adaptation to context. | Transition from regional to national coverage. |
| Organisational | Hospital care. | All healthcare settings. | All healthcare settings. | Health and social care. | All healthcare settings; health and disability care. |
| Overlap with quality management/systems | SMS separate from the quality management and occupational health and safety systems. | Quality and safety are considered together. | Quality and safety are considered together. | Safety is considered part of other systems in the organisation (eg, quality improvement). | Quality and safety are considered together. |
| Extent of accountability for 'other' organisations | Third-party management (outsiders performing activities in hospital) included. | – | – | There should be formalised agreements in place for the provision of services provided externally, including quality and safety. | – |

SMS, safety management systems.

**Table 4** Comparison of key features of the safety management systems or safety approach in each country relating to leadership commitment and safety policy and safety risk management

|  | Netherlands | Australia | Canada | Ireland | New Zealand |
|---|---|---|---|---|---|
| Leadership commitment and safety policy | BoD formulates multiyear safety policy and safety plan using SMART-formulated objectives; provides resources (funds, time, expertise, training, etc); reviews and evaluates impact and promotes safety culture; management translates the above into their own area of responsibility and reports to BoD. All employees have responsibility and are encouraged to participate in improving safety. Patient safety is included in the hospital's performance system | The Standards state that the governing body should provide leadership, set the strategic direction (including a framework with processes to drive improvement) and monitor and review safety and quality. The governing body of the organisation should also ensure 'that roles and responsibilities are clearly defined for the governing body, management, clinicians and the workforce' (p. 6)[28] in relation to quality and safety and that clinicians understand their role in patient safety. | 'Board members are accountable for safety and for ensuring patient safety concerns and incidents are acted upon' (p. 42)[29]; this includes quality and safety goals. Health leaders should encourage and support health teams to improve patient safety, while health workers should take an active role in creating a safe environment. | The national standards set out the need for 'A commitment to quality and safety articulated and demonstrated by those governing and leading the service' (p.72).[33] There should be an identified individual with accountability and responsibility for safety, as well as a commitment to safety from leaders at all levels. The national patient safety strategy will 'put in place integrated governance structures with clear accountability for planning, managing and addressing the above patient safety priorities' (p. 17).[35] That is, boards, management, staff and relevant quality and safety committees within an organisation should all have defined responsibilities for safety. | Leadership is one of the system drivers in the draft Clinical Governance framework. It enables the health workforce to be active members of the learning organisation by creating a culture of psychological safety and trust. Organisations need to share leadership by valuing knowledge and expertise drawn from lived experience of consumer and whānau equally alongside clinical and other knowledge. The National Adverse Events Policy states that health and disability service providers are accountable for ensuring the systems they implement recognise various degrees and types of harm; meet the national expectations for reporting, healing, learning and improving; involve consumers and whānau and create local processes to operationalise the policy. |
| Safety risk management* | Prospective risk assessment carried out by multidisciplinary teams using accepted methods (eg, FMECA - failure modes, effects and criticality analysis, which is used to identify potential failures and their severity within a system), provided with sufficient time and resources; regularly reviewed. Operational control measures: patient safety is integrated in daily business operations; management of change is an aspect of this. | Providers should identify and document risks (including external and internal disasters and emergencies); take actions to reduce them; regularly review actions and report on identified risks to staff and patients. | – | The national patient safety strategy details the need to 'proactively identify areas of risk to patient safety' (p. 15),[35] as do the Standards. Learning should also be taken from where things have gone wrong. Governance for risk should be integrated into an organisation's management processes. | In the draft Clinical Governance framework, the focus is on system safety and 'learning together' culture. It suggests using the Measurement and Monitoring of Safety Framework and actively seeking organisational data that challenges the sense that 'all is well'. A broad range of data sources are listed including quality monitoring, 'soft intelligence', operational data, health workforce and consumer and whānau experience. |

Continued

**Table 4** Continued

| | Netherlands | Australia | Canada | Ireland | New Zealand |
|---|---|---|---|---|---|
| | Focus on 10 priority themes. | The National Safety and Quality Health Service Standards (six of which relate to high-risk areas of patient care) and clinical care standards. | | Clinical standards. | Clinical standards. |

*Including risk related to planned changes.
BoD, board of directors.

searches. As the number of studies found was too large to allow a meaningful synthesis, we classified this evidence on impact and implementation according to whether they were 'rich' or 'thin', with 'rich' studies considered to be those with conceptual information which would allow us to answer the review's objectives and where the study data were close to the review objective.[20] 'Thin' studies did not contain conceptually rich data, and/or their objectives were not aligned with those of the review.[20] Our analysis concentrates on the rich evidence due to our focus on the implementation of patient safety approaches nationally and the need, therefore, for detailed, explanatory accounts of implementation at this level. Thin studies were included to illustrate the way in which a programme was designed or implemented and not because of their specific results, as they did not provide the necessary conceptual richness, tending to focus on settings and outcomes that were not generalisable (eg, specific organisations, small-scale interventions or outcomes less relevant to our research questions, such as the accuracy of a tool).

A case study approach was taken to the analysis of policy documents and other evidence to fully describe and provide contextual information on the patient safety approach in each country, enabling understanding of how these approaches related to the components of an SMS and cross-country comparison.[19 21] Each country's approach was described and analysed separately, and draft descriptions for each country were shared with experts from patient safety organisations in the respective countries for correction or comment. In the final step of the analysis, we compared the experience and implementation of the SMS or patient safety approach across countries. This involved the development of the initial framework—the four main components of an SMS remained the same, but subcategories were revised based on the information in the included evidence. Included publications were not quality appraised, as the majority were not empirical studies, so quality appraisal would not have made a significant contribution to our understanding of the application of SMSs in healthcare. We did consider uncertainties in the evidence, stemming from methodological limitations, when reporting on the Netherlands (as the only country that had explicitly used an SMS approach).

## Public and patient involvement and engagement

Due to the focus of the review, we sought public collaborators who had some knowledge of the topic. We held meetings with three public collaborators, two of whom had knowledge of safety management in other industries (chemical and police/counter-terrorism) and one who had experienced a safety incident in the NHS and had been campaigning to raise awareness of patient safety. In these meetings, we discussed their experiences of safety in other industries (where relevant) and of patient safety in the NHS, as well as their views on the improvement of patient safety. Our public collaborators provided important contextual information on safety management and SMS in other industries (eg, the importance of clear accountability for safety) and their potential use in the NHS. They also informed specific aspects of the review (eg, discussions of their own safety experiences in the NHS highlighted areas to consider in our data extraction and analysis, such as approaches to involving patients in safety policy and incident investigations and applying learning from safety incidents).

## RESULTS

Database searches resulted in 11 532 hits; 6913 records were screened at the title and abstract level. Of those, 158 were selected for full-text screening, and 89 were found to meet the inclusion criteria. Searches of websites resulted in 32 publications being included in the review, along with 22 documents from other sources (14 from the reference lists of included documents and eight from expert suggestions). This resulted in a total of 143 studies being included in the review. Further details can be found in figure 1 and online supplemental file 1.

Due to the quantity of evidence found, we grouped publications into two broad categories (table 2):

► Descriptions of each country's approach to patient safety, including influences from high-risk industries (mainly policy documents but also outputs, eg, opinion papers).
► Evidence on the impact and implementation of a specific patient safety approach or aspects of it (mainly research papers and project reports). As described above, these were categorised further as:
1. 'Rich': reporting on the impact or implementation of the approach, or an essential part of it.

**Table 5** Comparison of key features of the safety management systems or safety approach in each country relating to safety assurance (including subcategory on incident reporting and investigation) and safety promotion and culture

| | Netherlands | Australia | Canada | Ireland | New Zealand |
|---|---|---|---|---|---|
| **Safety assurance*** | | | | | |
| Safety performance monitoring and measurement | Management is responsible for monitoring patient safety performance in the hospital and reporting of results (eg, to BoD) at least annually. | Organisations should have quality improvement systems that identify, monitor and report on safety performance and outcomes | 'Ensure that avoidable death and hospital standardised mortality ratio data is tracked and reported so action can be taken to improve outcomes' (p. 34).[29] | Monitoring against National Standards for Safer Better Healthcare by Health Information and Quality Authority, using national indicators where possible. | Monitoring against Te Tāhū Hauora's health quality and safety indicators. The Quality and Safety Measures are a mix of process and outcome measures and focus on key safety priorities: falls, healthcare associated infections, surgical harms and medication safety. |
| Continuous improvement | Using a multidisciplinary approach, improvement measures are implemented and their impact assessed. | Organisations should identify areas for improvement and implement and monitor improvement strategies. | 'Support an audit and feedback system to ensure that recommendations to improve patient safety are implemented' (p. 33).[29] | Requirement for annual audits, national and local. Learning from both incident reporting and monitoring should be used to improve quality and safety. | Continuous learning from adverse events is based on open reporting, investigation and meaningful analysis. Changes are implemented based on the learning process. |
| **Incident reporting and investigation** | | | | | |
| Reporting system | Requires safe, efficient and effective reporting processes and a system using accepted methods and prespecified incident categories (reporting limited to hospital and to BoD). The information is used to improve safety by determining tasks, powers and responsibilities. | National Safety and Quality Health Service Standards require an organisation-wide clinical incident management and investigation system. | Health leaders should enable health teams to 'Report, learn from, and act on patient safety concerns and incidents' (p. 24)[29] by ensuring policies and reporting mechanisms are in place. Organisational incident reporting and reporting to national systems for certain incidents. | The Standards detail the need for 'Arrangements to identify, manage, respond to and report patient-safety incidents in a timely manner'.(p. 66).[33] Incidents need to be entered on a national system, whether they are investigated internally or externally depends on severity of incident. | The National Adverse Events policy 2023 requires serious adverse events to be reported to and investigated by the Te Tāhū Hauora. Less serious events are reported and managed locally. |
| Method of investigation | Root cause analysis was recommended as an investigation method initially.[25] NEN8009 (2018) does not recommend a method.[26] | Investigation method is root cause analysis. | – | Investigation method based on systems analysis.[36] | Investigation method is learning review. |
| Postincident support | Staff involved in an incident are offered aftercare. | – | Health leaders and boards should ensure that health teams have access to psychological support. | Support (including psychological) for staff involved in a serious incident. | A theme in The National Adverse events policy 2023 'healing': providing emotional and practical support to affected individuals, families and healthcare workers. |
| **Safety promotion and culture*** | | | | | |
| Presence of safety culture | BoD responsible for promoting safety culture, including safe reporting. Open communication in all directions. | Organisations should provide 'timely reports on safety and quality systems and performance' (p. 7)[28] to employees at all levels of the organisation, the public and other healthcare organisations. | Recent move to create a proactive environment, with a culture and processes that support continuous safety improvement (eg, safety huddles and other methods of open communication). | Systems for staff communication and engagement within organisations as well as with external agencies (eg, Health Information and Quality Authority) and support for programmes to promote a patient safety culture. | Cultural safety; integration of Maori worldview and safety science. |

Continued

de Bell S, *et al. BMJ Open* 2026;**16**:e107772. doi:10.1136/bmjopen-2025-107772

**Table 5**  Continued

|  | Netherlands | Australia | Canada | Ireland | New Zealand |
|---|---|---|---|---|---|
| Training and education | BoD and management are responsible for providing training and monitoring whether staff need further training. | The organisation should provide access to training and monitor participation. | Health teams have a responsibility to undertake training and leaders and boards to provide training. | The Strategy identifies the importance 'designing and delivering safety information and training' (p. 13)[35] for staff. | Staff should be provided with learning opportunities. |
| Patient participation | Patient participation in drawing up and implementing safety policy and reporting incidents, complaints and claims. | Patient participation in 'healthcare governance planning, design, measurement and evaluation' (p. 19)[28] | Patient participation in patient safety (eg, through incident reporting and as public representatives in policy discussions). | Organisations should 'develop mechanisms to empower patients to contribute' (p. 11),[35] including to governance structures and reporting incidents. | 'Organisational clinical priorities, processes and evaluations are co-designed and developed collectively' (p. 11)[58] with patients. |

*Including accreditation, monitoring and reporting of outcomes as well as systematic implementation of improvement measures and evaluation of their impact, including patient and family involvement.
BoD, board of directors.

2. 'Thin': papers with a very narrow focus and limited relevance to our review question (eg, the accuracy or acceptability of a risk assessment tool implemented as part of a patient safety programme).

Our analysis was based primarily on the descriptive and rich evidence. We focused on 53 publications, which can be found in online supplemental file 1, from Australia (5), Canada (7), Ireland (8), New Zealand (9) and the Netherlands (24) (table 1). Some 'thin' evidence is used to illustrate specific points (see full list in online supplemental file 1). Below, we describe the patient safety approach in each country and then compare these approaches.

### Netherlands

The Netherlands was the only country to explicitly use an SMS approach, in the form of its national patient safety programme (PSP), which was launched in 2008 and ran until the end of 2012.[22] The programme was implemented in all public hospitals and was based on the premise that change in the structure or processes of an organisation is necessary to see a change in outcomes.[23] It had two pillars: (1) hospitals were required to implement a certified SMS and (2) a set of guidelines focusing on improving patient safety for 10 priority themes.[13] The aim was a 50% reduction in adverse events and potentially preventable deaths over a period of 5 years.[13] The two pillars of the PSP continued to be implemented after 2012, with development and revisions based on evaluations of its success. A new safety programme, based on a Safety-II perspective (see the Glossary section), began in 2020.[24]

The basic SMS requirements were agreed by key stakeholders and published in 2007 (NTA 8009).[25] The complexity of implementing an SMS was acknowledged, with healthcare providers required to carry this out in stages. In Phase I, the basic requirements of an SMS were developed (leadership, management, personnel, patient participation, prospective risk assessment, retrospective risk assessment and improvements), with further requirements added in Phase II (communication,

third-party management and control measures).[25 26] The latest version of the document specifying these requirements, NEN 8009, was published in 2019; it introduces a new perspective on patient safety based on Safety-II and resilience engineering (see the Glossary section). These concepts emphasise the importance of learning from successful, 'incident-free' practice as well as from the investigation and prevention of incidents.[26]

There were 10 high-priority themes for patient safety, such as patient mix-up (see the Glossary section). Expert groups were convened for each theme, and guides were developed describing structure, processes and outcomes. Interventions to improve patient safety were explained during national theme conferences. The programme also had a website where resources and patient-facing materials were available. While the initial focus was on adult care, six of the patient safety themes were later expanded to include paediatric hospital care.

### Australia

In Australia, patient safety is primarily a national (ie, federal government) responsibility.[27] However, state and territory governments are responsible for their own public hospital systems and the licensing and regulation of the private hospital sector, as recognised in the National Health Reform Act 2011 (which specifies responsibilities for funding, operating, managing and regulating the health system).[27] The systems and processes used to ensure safety therefore differ between territories and states.[27]

The Australian Commission for Safety and Quality in Healthcare developed the National Safety and Quality Health Service (NSQHS) Standards with the aim of ensuring the relevant systems are in place within a healthcare organisation to ensure quality of care.[28] The first edition was published in 2012, the latest in 2021.[28] The NSQHS Standards detail requirements for healthcare organisations regarding clinical governance and partnering with consumers, as well as specific areas of clinical management. They require providers to have

an organisation-wide clinical incident management and investigation system, with an open disclosure policy consistent with the Australian Open Disclosure Framework.[27] Certain indicators, which should be measured, such as sentinel events, have been developed by the Commission, and the Australian Health Performance Framework details a common set of metrics for public reporting of safety and quality data.

Australia has not used an SMS approach in its healthcare system, although '*the healthcare sector began to draw on lessons from other high-risk industries*' such as aviation and mining, as well as using human factors analysis, to develop its patient safety approach.[27] These concepts from high-risk industries have influenced specific elements of Australia's approach (eg, surgical checklists).

### Canada

The responsibility for delivering health services lies with the provincial and territorial governments, leading to variation in how patient safety is approached in Canada. The Canadian Quality and Patient Safety Framework was launched in 2020 and provides guidance to drive patient safety and quality improvement in all healthcare sectors and settings. While it is intended to align Canadian legislation, regulations, standards, organisational policies and public engagement nationally,[29] there is currently variation between governments. Not all jurisdictions have quality and safety plans or policies regarding key elements of patient safety.[30]

Healthcare Excellence Canada, an independent non-profit organisation funded primarily by Health Canada—a department of the federal government—has produced a discussion guide that details the recent change in Canada's approach to patient safety. There has been a move from 'traditional' patient safety improvement (eg, scorecards and audits) to a focus on creating a proactive and inclusive environment, with a culture and processes that support continuous safety improvement.[31] Canada has drawn on learning from high-risk industries, with the Measurement and Monitoring of Safety Framework (MMSF) underpinning its new approach to patient safety but has not taken an SMS approach.[32]

### Ireland

The Health Information and Quality Authority developed the National Standards for Safer Better Healthcare in 2012; these detail expectations for healthcare and guide improvement of quality, safety and reliability.[33] They were replaced by a principles-based approach in 2021.[34] However, the Health Service Executive has developed a national patient safety strategy which builds on the standards, to be implemented and guide improvement at a local level.[35]

Irish healthcare providers are required to have an incident reporting system.[36] Incidents are entered into the national system and reviewed externally or internally, depending on the severity of the incident.[36 37] Providers are also expected to undertake clinical and non-clinical audits

(which may be part of national or local programmes), implement improvements based on these[35 38] and assess and manage risks, with a formal process in place to do this.[39] Sláintecare, an improvement plan and strategy intended to reform and develop health services in Ireland over the next decade and into the future, is currently being implemented in Ireland; the provision of healthcare, including the improvement of safety, is consequently evolving.[35]

While Ireland has not taken an SMS approach, the influence of high-risk industries can be seen in various elements of its approach. The national patient safety strategy details an action regarding information and training for staff drawing on 'patient safety and reliability science, systems thinking, audit, quality improvement methodologies, change management, human factors and multidisciplinary team working for safety' (p. 13),[35] with an example of this being the delivery of simulation-based education and human factors training to improve patient safety (see the Glossary section).[40 41]

### New Zealand

Patient safety in New Zealand is a central focus of the healthcare system, with the Te Tāhū Hauora Health Quality and Safety Commission (Te Tāhū Hauora), established in 2010, playing a pivotal role in improving healthcare quality and safety across the country.[42] The Pae Ora (Healthy Futures) Act 2022 replaced the country's 20 District Health Boards, aiming to create a more centralised, effective and efficient healthcare system.[42] The Act[42] and the updated Nga Paerewa Standard Health and Disability Services Standard[43] are associated with important changes in New Zealand's patient safety policy, such as the 'Healing, Learning and Improving from Harm: National Adverse Events Policy',[44] which took effect from 1 July 2023 and the new 'Clinical Governance Framework: Collaborating for quality' (which is still in a process of consultation).[45]

The new framework has been intentionally designed for the cultural context of New Zealand, with service users and their families seen as 'partners in care'.[45 46] Elements of safety are emphasised, such as the importance of having processes and systems in place to ensure quality and safety (eg, clinical risk management systems to deal with non-compliance with clinical standards and policies) and of focusing on 'system safety'—managing risk within a systems perspective—and learning how the healthcare system shapes the experiences of consumers and staff.[45] The MMSF is suggested as a tool that could help decision-makers to assess the overall safety of their organisation. To assess the quality of the health and disability system, Te Tāhū Hauora has adopted a range of baseline measures and indicators:

► Atlas of Healthcare Variation.
► Quality and Safety Markers.
► Quality Accounts.
► Health Quality and Safety Indicators.
► Prevention quality indicators.

The current (2023) version of the clinical governance framework (in consultation) includes a number of ideas (eg, systems safety and resilient organisations) which are linked to developments in high-risk industries.[45 47] For example, the 'learning review' method was originally developed for the investigation of firefighter incidents in the context of the US Forest Service and draws on a number of concepts, such as human factors and ergonomics, systems safety and resilient organisations. It was specifically developed for 'adaptive complex systems'—complex systems with a high level of uncertainty that interact, learn and adapt over time—such as healthcare.[47]

## Comparison

Although the Netherlands was the only country to explicitly use an SMS approach, the main components of an SMS were identified, to varying extents, in the patient safety policies and initiatives of the other four countries, as described in tables 3–5. While the interdependence of these elements was acknowledged, they were often detailed in different policies and within the scope of different organisations. Key features that were seen in all five countries, organised according to the components of an SMS, included the following.

### Leadership commitment and safety policy
► Leadership commitment to patient safety.
► National patient safety policy (separately or in some cases as an aspect of wider quality improvement policy).

### Safety risk management
► Prospective risk management, with risks identified using multiple sources of evidence (eg, local, national and international research). Countries had specific standards for high-priority patient safety topics (eg, patient identity mix-up) as well as general guidelines.

### Safety assurance
► Retrospective incident reporting and analysis linked to a system for feedback and learning to ensure continuous improvement. Methods for categorising events (eg, actual harm vs near misses) and their severity (high and low) were similar across countries and determined whether and how they were investigated (eg, central vs local, such as in the Netherlands and New Zealand). Investigations differed (eg, learning review and restorative practice in New Zealand; root cause analysis in the Netherlands and Australia; see the Glossary section).
► Monitoring of patient safety performance. While systems for monitoring at the local level were less clear, all countries had a national system.

### Safety promotion and culture
► Focus on developing a safety culture (including open communication and blame-free reporting).

► Patients' and families' involvement, both in the development of patient safety policies and in safety at the local level.

There were, however, differences in the implementation of these common features between countries (eg, incident investigation as described above and see table 5). There were also other areas of variation, particularly relating to the scope of the patient safety approach (table 3). In most cases, patient safety was considered alongside healthcare quality improvement, although in the Netherlands, it was separate. Implementation of policies varied from national to regional, and their application to all or specific healthcare settings (eg, hospital care only in the Netherlands or all care settings in the other four countries). Additionally, a range of concepts from high-risk industries and safety science (see the Glossary section), more broadly (eg, Safety II, human factors and resilience engineering; see the Glossary section) had influenced the patient safety approaches of different countries.

All countries considered inequalities to some extent by targeting groups particularly at risk of unsafe care (eg, older people). The most specific guidance was found in Australia, Canada and New Zealand. In these countries, groups were identified in terms of social (eg, those living in deprivation), ethnic (eg, Māori and Pacific peoples in New Zealand) and protected characteristics (eg, sexuality) and actions or objectives were specified to achieve outcomes for these groups. Ireland highlighted the need to consider a person's circumstances and background in their healthcare; the PSP in the Netherlands did not provide guidance for different patient groups beyond the need to pay special attention to older people (eg, within the 10 patient safety themes).

It is difficult to directly attribute changes in patient outcomes to specific policy and management changes, especially when different components are introduced incrementally and often over many years. In relation to our third objective, on the effectiveness, implementation or experience of SMSs or patient safety approaches, we found that only three countries conducted national evaluations of their approach. In the Netherlands, a longitudinal study comparing data from 2008[48] and 2011/2012[49] reported a 45% decrease in potentially preventable adverse events over the course of the national safety programme (which required all public hospitals to implement an SMS and 10 safety themes). However, a more moderate 30% decrease in potentially preventable adverse events, which was not statistically significant, was found after adjusting for data clustering and changes in the patient mix between measurements.[50] There was also variation in the successful implementation of the 10 themes between hospitals and departments, with a number of staff-related, organisational and topic-specific factors identified as potential modifiers.[23] Australia and Ireland both found improvements for certain aspects of patient safety,[27 51 52] with some of these in areas in which the Netherlands also saw improvement (eg, medication safety).[13] Similarly, there were factors which all of these

countries identified as facilitating better patient safety, such as strong governance or leadership.[13 27 51 52]

## DISCUSSION

Of the five included countries, only the Netherlands has formally adopted an SMS approach.[13] All SMS components were found in the policies of the other four countries, but not in the form of a single unified system. Across all five countries, there was a change in the patient safety discourse away from a narrow focus on reporting and learning from incidents. While reporting was still considered important, the new approaches to patient safety adopted broader definitions of safety (eg, including psychological and cultural safety) and harm (eg, including harm resulting from social inequalities and structural oppression) and emphasised taking a systems perspective to safety, learning from successful practice as well as failures. They highlighted the need for wider involvement in the processes of assessing and creating safety, including patients and families as well as staff.

Healthcare is often thought to have more complex and unpredictable demands compared with other industries, which have adopted an SMS approach, leading to debate over the suitability of SMSs in contexts and organisations in which service delivery is not stable or predictable.[5] The Dutch patient safety programme had some positive impact on patient safety, suggesting an SMS approach is suitable for use in healthcare.[13] However, the evaluation focused on the safety programme as a whole, rather than the SMS specifically, making it difficult to determine whether there are specific components of an SMS that are particularly effective in improving safety. While improvement was seen as part of the programme, this does not mean it is the only way to improve patient safety.[5] The elements of an SMS were present, to some extent, in the healthcare systems of all included countries, but other concepts used by high-risk industries and from safety science more widely had influenced the patient safety approach in all countries, with evidence of effectiveness in some cases (eg, Ireland).[51]

The use of these concepts potentially reflects a new, emerging perspective in which healthcare and high-risk industries are seen as societal and corporate domains that share the same theoretical field—safety science—and can learn from each other, as recognised in documents such as the WHO Global Patient Safety Action Plan 2021–2030.[1] This is very different from the idea that, in terms of safety, healthcare is lagging behind and should learn and even copy the high-risk industries' approach to process safety, which has dominated the patient safety discourse.[5 53] There were considerable differences in the implementation of SMS elements and safety concepts between countries, with patient safety systems reflecting different cultural and organisational realities and local preferences. What works, or makes sense, in one context may not work, or make sense, in another. Comparing and understanding such differences could advance the

field of safety science as applied to healthcare and help countries review and improve their own policies and practices. For example, a national review of patient safety in England in 2024 found that, while individual healthcare organisations are prioritising common patient safety problems, there is a need for a more focused and coherent approach at the national level.[2] The experiences of a smaller country, such as the Netherlands, which was able to implement a single, highly structured patient safety programme, may be more applicable to the UK context than those of larger, federal countries such as Australia and Canada.

Within this general shift in thinking, different countries had a preference for specific concepts and theories from safety science, the most prominent of which were the MMSF and resilience engineering theory (see the Glossary section). Regardless of the theoretical underpinnings of a particular patient safety approach, it still needs to be both operationalised and adapted to a particular context. While we focused in this review on how a national organisation can influence or coordinate the implementation of an SMS or safety approach, some policies and most interventions are implemented at the level of service commissioning and/or within health service delivery organisations. For example, the MMSF is a framework and a set of 'guiding principles for safety measurement and monitoring' (p. 70).[5] It does not provide details on implementation, which remains a challenge, as can be seen in the experience of Canadian healthcare providers in applying the MMSF in their healthcare system.[32] In terms of adaptation, current patient safety discourse recognises that both 'Safety I' (learning from incidents) and 'Safety II' (learning from success) are important and should be part of a modern patient safety approach. Safety II has important implications for the organisation of patient safety, particularly the use of a broader range of safety metrics, including outcome measures and resilience indicators, as well as adverse events.[54]

However, chosen measures and indicators need to be suitable for the context in which they are being used. As monitoring and evaluation are integral to a systems approach to safety, this is a key area for further research. Realist research, to understand the causal mechanisms through which patient safety initiatives work and contextual factors that influence their successful implementation, could aid the choice of suitable methods of evaluation. Understanding the mechanisms behind different components of patient safety initiatives would also support their adaptation to different contexts. For example, the dominant focus of patient safety approaches in all countries was on hospital care (although all countries apart from the Netherlands included all healthcare settings in their patient safety approaches), but there may be different challenges in other settings, such as primary and community care, and for specific patient groups (eg, patients with learning disabilities), requiring the tailoring of standards.

## Limitations

Although we followed best practice for conducting a systematic review, some limitations should be acknowledged:

► The review is focused on patient safety in selected high-income, English-speaking countries with healthcare systems broadly comparable to that in the UK. Although we conducted scoping searches to decide on our countries of focus, this means that there may be different contexts (eg, lower-income countries) in which SMS are being implemented within the healthcare system.

► Our interpretation is based on analysis of non-evaluative and administrative documents, many of which express aspirations regarding patient safety, rather than being empirically grounded evidence detailing outcomes. We tried to mitigate this limitation by talking to experts from each country and sharing our draft descriptions with them for comment.

## CONCLUSIONS

Only the Dutch patient safety programme was explicitly based on an SMS approach and showed some longitudinal evidence of improving patient safety. Process safety, as conceptualised in high-risk industries, appeared to have been adapted or had some influence on approaches in the other countries, but this was less systematic and explicit. Approaches to patient safety in all countries also draw on a range of concepts from broader safety science. This suggests a shift from the view that healthcare needs to adopt high-risk industries' approach to safety and increasing awareness that for initiatives to be successful, they need to be adapted to their context, especially through more collaborative learning between industries and sectors.

## GLOSSARY

► *Human factors* is the 'study of the interrelationships between humans, the tools they use, and the environment in which they live and work' (p. 210).[54]

► *Learning review* is a specific approach for investigating incidents where the focus is on collaboration and learning and the system rather than on the individuals involved.[55]

► *Patient mix-up*: Errors regarding a patient's identity or their procedure (eg, surgery involving the wrong side, wrong site, wrong procedure or wrong patient).[13]

► *Patient safety*: 'a framework for organised activities that creates cultures, processes, procedures, behaviours, technologies and environments in healthcare that consistently and sustainably lower risk, reduce the occurrence of avoidable harm, make errors less likely and reduce the impact of harm when it does occur'.[1]

► *Resilience engineering theory* 'looks for ways to enhance the ability of systems to succeed under varying conditions', that is, proactively building resilient systems.[5 46]

► *Resilient systems/organisations* can adjust their functioning before, during or after events, sustaining required operations under both expected and unexpected conditions.[46]

► *Restorative practice* in the context of the New Zealand patient safety policy '…is a voluntary, relational process where skilled facilitators support all those affected by an adverse event in a safe and supportive environment. Participants can speak openly about what happened to understand the human impacts and to clarify responsibility for the actions required for healing and learning'.[56]

► *Root cause analysis* is a process used to identify the root—or most fundamental—cause of a problem or incident (rather than the first viable cause).[6 7]

► *Safety I* is an approach to safety in which the focus is on preventing errors and incidents from occurring by identifying the causes and responding either when an event occurs or when a risk is deemed unacceptable.[55]

► *Safety II* is an approach to safety which shifts the focus from what goes wrong to ensuring that things go right (learning from success).[5 57]

► *Safety science* is considered to be any research, education, etc, concerning the science and technology of human safety.[58]

► *SMART* stands for Specific, Measurable, Acceptable, Realistic and Time-bound, in relation to setting objectives.

► *System safety* acknowledges that multiple factors and the interactions between them lead to safety and that the system can be modified to improve safety and resilience.[5]

► *A systems approach* considers multiple levels, that is, 'the interaction between individuals, organisations and sociotechnical systems' (p. 25).[1]

► *Wrong-site surgery*: a surgical procedure carried out on the wrong side of the body or on the wrong site on the body.[13]

**Acknowledgements** The Exeter HSDR Evidence Synthesis Centre team would like to acknowledge and thank Tracey Herlihey, Matt Fogarty and Jason Cox for their support in developing this review, the experts who took the time to provide us with insight on patient safety in their countries, and our public collaborators for their valuable contributions to our understanding of the topic. We would also like to thank Sue Whiffin and Jenny Lowe for administrative support throughout this review.

**Contributors** Conceptualisation, writing, editing and reviewing: SdB, ZZ, AB, JTC and RA. Formal analysis and writing the original draft: SdB and ZZ. Funding acquisition and project administration: JTC and RA. Investigation: SdB, ZZ, AA and MC. SdB is the guarantor.

**Funding** This report presents independent research funded by the National Institute for Health Research (NIHR). This review (award NIHR136105) was commissioned by the Health Services and Delivery Research (HS&DR) programmes as part of a series of evidence syntheses under award NIHR130538. For more information, visit https://fundingawards.nihr.ac.uk/award/NIHR135660 and https://fundingawards.nihr.ac.uk/award/NIHR130538. JTC and AB are also supported by the National Institute for Health Applied Research Collaboration South West Peninsula.

**Competing interests** SdB, ZZ, AB, MC and RA have no competing interests. JTC: Member of the NIHR HTA General Board (2019-2024), member of the NIHR PRU Commissioning Panel (2022), and member of the NIHR RPSC Committee (2024).

**Patient and public involvement** Patients and/or the public were involved in the design, or conduct, or reporting, or dissemination plans of this research. Refer to the Methods section for further details.

**Patient consent for publication** Not applicable.

**Ethics approval** Not applicable.

**Provenance and peer review** Not commissioned; externally peer reviewed.

**Data availability statement** Data sharing not applicable as no datasets generated and/or analysed for this study. This is an evidence synthesis study based on published primary research, it did not generate new data. All data extracted from the included publications, along with links to each publication, can be found in the report or Appendices. Further information can be obtained from the corresponding author.

**ORCID iDs**
Siân de Bell https://orcid.org/0000-0001-7356-3849
Zhivko Zhelev https://orcid.org/0000-0002-0106-2401
Jo Thompson Coon https://orcid.org/0000-0002-5161-0234

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
