## [Reviewer comments · BMJ Open]

ARTICLE DETAILS

Title (Provisional)

The implementation of Safety Management Systems (SMS) in healthcare: a systematic review and international comparison

Authors

de Bell, Siân; Zhelev, Zhivko; Bethel, Alison; Clarke, Maria; Anderson, Rob; Thompson Coon, Jo

VERSION 1 - REVIEW

Reviewer	1
Name	Koike, Daisuke
Affiliation	Fujita Health University School of Medicine, Department of Public Health
Date	15-Aug-2025
COI	None

Reviewer Comments

Thank you for the opportunity to review the manuscript entitled “The implementation of Safety Management Systems (SMS) in healthcare: a systematic review and international comparison.”

This study investigated SMS across five predefined high-income countries. The healthcare systems varied significantly across the included countries/regions. Various types of SMS were identified through this review, and the authors summarised these systems into four components: Leadership commitment and safety policy, Safety risk management, Safety assurance, and Safety promotion and culture. However, due to the unclear study aim, incomplete description of the methods, and insufficiently detailed results, the manuscript currently offers limited value to readers.

General comments

- Need for a more focused aim and alignment of results

The aim of the study should be stated more clearly and with a sharper focus, and the results should be restructured to align closely with that aim. To ensure the paper has relevance beyond a UK audience, the search scope should be expanded beyond countries already known to have a national healthcare system, so that the findings are meaningful to readers from other countries as well.

- Unclear study focus

The study's aim and focus are unclear. The safety management system is not clearly defined, making it difficult to understand exactly what was investigated and the overall value of the manuscript. The evidence gap is also not clearly articulated; while the authors describe SMS in other industries, they do not specify the evidence base regarding other countries or the current status of healthcare in any country. The rationale for why healthcare should implement SMS from other industries is unclear, and given that industries often influence each other, such implementation is not surprising. The Introduction should clearly define what is already known and what is unknown.

- The authors state, "How are the components of an SMS reflected in the healthcare policy documents?" This is not a suitable question for a systematic review, as it focuses only on nationwide healthcare policy documents.
- The question, "What does evidence from beyond the UK tell us about the effectiveness, implementation or experience of SMSs within healthcare?" appears targeted at a UK readership. If this is the primary audience, a local journal may be more appropriate. BMJ Open is an international high-impact journal.

Methods

- No inclusion or exclusion criteria, and no search query, are provided in the manuscript. Even if these are described in the supplementary materials, the main criteria should be included in the manuscript.
- Please justify the decision to focus on "rich evidence" and to exclude "thin evidence."
- The exclusion of the UK seems inappropriate for a systematic review, as the UK has a national healthcare system and is a high-income country. Other examples of high-income countries with a national healthcare system include Spain (SNS: Sistema Nacional de Salud), Hong Kong (Special Administrative Region of China), and Japan. The rationale for restricting the analysis to only certain countries should be fully explained, and the validity of applying an income-based restriction should also be justified.
- The analytical framework for the review is not described in the method.

Results

- Please summarise the results according to the framework: “Leadership commitment and safety policy,” “Safety risk management,” “Safety assurance,” and “Safety promotion and culture.” If any other results were extracted from the review, please describe them explicitly as well.
- The authors state, “The Netherlands was the only country to explicitly use an SMS approach.” However, the evidence for this statement is not clear; for example, Australia has the NSQHS. Why is this not considered an SMS approach?
- In Table 4, the meaning of the rows is unclear. For example, “Safety assurance” includes two rows, and “Incident reporting and investigation” has three. It would be helpful to include sub-section labels for each row so that it is clear what is being described.
- The results of the analytical framework—examining the components of an SMS by reviewing policy documents and review articles from high-risk industries—are not clearly presented.

Discussion

- The first paragraph of the Discussion does not appear to be based on the results.
- The second paragraph primarily restates the findings rather than interpreting them. The Discussion would be strengthened by elaborating on the implications of these findings—for example, identifying which components of an SMS approach may be most effective in healthcare, and clarifying both the observed impacts of SMS and the anticipated impacts that such an approach aims to achieve.
- The third paragraph introduces a new perspective in which healthcare and high-risk industries are viewed as sharing the same theoretical field—safety science—and able to learn from each other. However, no evidence or examples are provided to substantiate this claim. Furthermore, if the relationship is truly reciprocal, this would imply that there are findings or practices from healthcare that could be transferred to high-risk industries, but such examples are not presented. Including concrete evidence or illustrative cases would strengthen the argument.
- The fourth paragraph does not clearly articulate interpretations derived from the study’s findings. This section should explicitly base the discussion on the results of this review and develop arguments that directly emerge from those findings.
- In the Discussion, it is important to go beyond reiterating general principles and instead interpret the findings in light of the study’s specific results. The most compelling discussion sections explain how the evidence obtained in the present review confirms, contradicts, or extends existing knowledge, and what implications arise from these findings for policy, practice, or future research.

Conclusions

- Overinterpretation of SMS impact

SMS-based activities are implemented at the level of individual facilities; therefore, their outcomes should be evaluated at the facility level rather than the national level. Concluding that the SMS approach had an impact only in the Netherlands appears to overinterpret the evidence.

- Causal inference regarding process safety from other industries

The statement that process safety concepts from other industries influenced the approaches in other countries implies a causal relationship not supported by the evidence. A more appropriate description might be that the frameworks or ways of thinking from process safety in other industries can be adapted to healthcare, or that similar concepts have been applied in healthcare.

- Shift toward contextual adaptation and cross-sector learning

The assertion that there has been a shift toward recognising the need for contextual adaptation and collaborative learning across industries and sectors does not appear to be directly derived from the results of this study.

Technical terminology

Several key concepts are used without clear definitions—e.g., SMS, concepts from high-risk industries, safety science, and patient safety approach.

In this manuscript, “systems approach” appears to refer to the overall conceptual framework of system-oriented safety management, whereas “SMS” refers to the tangible system implemented in practice (e.g., manuals, procedures, reporting systems). However, in the context of patient safety, “systems approach” is commonly understood in contrast to the “person approach”—meaning a focus on organisational and process-level factors rather than individual blame, aiming to improve the systems and environments that prevent errors. To avoid confusion, I suggest clarifying the intended meaning of “systems approach” in this manuscript.

Questioning the Classification as a Systematic Review

While the manuscript is described as a systematic review, the inclusion of only five preselected countries and the use of a predetermined analytical framework raises concerns about whether it fully meets the standards of a systematic review as per PRISMA or Cochrane guidelines. Systematic reviews typically involve comprehensive and transparent searches without geographical restrictions unless these are clearly justified. In this case, the rationale for limiting the scope to these specific countries should be fully explained, and the

potential impact of this restriction on the generalisability of the findings should be discussed. The recommendation is to change this to a scoping review, or to expand the inclusion criteria for countries, or to limit the scope based on health insurance system type.

Reviewer	2
Name	Hirao, Tomohiro
Affiliation	Kagawa University Faculty of Medicine Graduate School of Medicine, Public Health
Date	16-Aug-2025
COI	None

The manuscript summarizes the application of SMS in five countries and is a useful for understanding the situation in developed countries.

There are no major comment with the content.

As this is a study commissioned by NHS England, it is understandable that it was written with the UK in mind. It would be helpful to include specific examples of where the findings could be applied.

P6L40, section 2.3 should be 2.4.

Reviewer	3
Name	Jarrar, Mu'taman
Affiliation	Al-Hussein Bin Talal University
Date	11-Nov-2025
COI	None

The paper contributes meaningfully to the theory of expert leadership and has practical implications for hospital management and leadership development. However, some sections could benefit from conceptual refinement, methodological clarification, reference update and a more critical interpretation of findings.

1. Authors need to build stronger case of study importance (stronger problem statement required).

2. Update references 2 and 5 regarding leadership commitment to safety, empowerment and the role of leadership. see the following references:

Hamdan, M., Jaaffar, A. H., Khraisat, O., Issa, M. R., & Jarrar, M. T. (2024). The association of transformational leadership on safety practices among nurses: the mediating role of patient safety culture. *Risk Management and Healthcare Policy*, 1687-1700.

Al-Bsheish, M., bin Mustafa, M., Ismail, M., Jarrar, M. T., Meri, A., & Dauwed, M. (2019). Perceived management commitment and psychological empowerment: A study of intensive care unit nurses' safety. *Safety Science*, 118, 632-640.

Al-Bsheish, M., Jarrar, M. T., Mustafa, M. B., Zubaidi, F., Ismail, M. A. B., Meri, A., & Dauwed, M. (2022). ICU nurses' safety performance related to respect for safety and management commitment: A cross-sectional study. *Contemporary nurse*, 58(5-6), 446-459.

Suggested Reference Integration

Reference to add:

Al-Bsheish, M., Jarrar, M. T., Al-Mugheed, K., Samarkandi, L., Zubaidi, F., Almahmoud, H., & Ashour, A. (2023). The association between workplace physical environment and nurses' safety compliance: A serial mediation of psychological and behavioral factors. *Heliyon*, 9(11).

This reference and the previously mentioned can strengthen your study case. In the Introduction or Discussion, where the manuscript discusses the mechanisms linking leadership or management practices with subordinate outcomes (job satisfaction, safety, or retention).

Specifically, it fits after line 155–160 of the Introduction, where the authors explain that “multiple factors influence job satisfaction and intention to stay, but middle managers play a particularly crucial role in maintaining satisfaction and reducing turnover intention.”

Suggestion to be considered:

“Beyond leadership style and HRM practices, the broader workplace environment and psychological conditions shaped by managerial behavior are also critical determinants of employee compliance, satisfaction, and retention. For instance, Al-Bsheish et al. (2023) demonstrated that supportive organizational environments and psychologically safe climates mediate the link between workplace factors and safety behavior among nurses, underscoring the complex behavioral pathways through which management practices influence outcomes.”

3. Consider deepening the discussion of mechanisms—specifically why HRM practices mediate the expertise–satisfaction relationship but perceived expertise does not. This contrast could be theorized further using social identity or legitimacy theories.

4. The data matching process between managers and subordinates is commendable but raises possible selection bias concerns due to the exclusion of unmatched cases. A short discussion or sensitivity analysis on how these exclusions might affect representativeness would strengthen the methodological transparency.

5. The claim that clinical expertise “causes” stronger HRM practices may be too strong for a cross-sectional design. Consider tempering causal language (e.g., “associated with” or “predicts”) throughout.

6. The practical recommendations are strong but could go further by addressing how organizations can develop HRM competence among expert clinicians—e.g., through mentoring, leadership coaching, or integrated management training within clinical pathways.

VERSION 1 - AUTHOR RESPONSE

Response to review

Comment	Response	Page no.
Reviewer 1		
Thank you for the opportunity to review the manuscript entitled “The implementation of Safety Management Systems (SMS) in healthcare: a systematic review and international comparison.” This study investigated SMS across five predefined high-income countries. The healthcare systems varied significantly across the included countries/regions. Various types of SMS were identified through this review, and the authors summarised these systems into four components: Leadership commitment and safety policy, Safety risk management, Safety assurance, and Safety promotion and culture. However, due to the unclear study aim, incomplete description of the methods, and insufficiently detailed		

results, the manuscript currently offers limited value to readers.		
General comments  • Need for a more focused aim and alignment of results The aim of the study should be stated more clearly and with a sharper focus, and the results should be restructured to align closely with that aim. To ensure the paper has relevance beyond a UK audience, the search scope should be expanded beyond countries already known to have a national healthcare system, so that the findings are meaningful to readers from other countries as well.	As this is a systematic review, it is not possible to change our focus or expand the scope of the review at this stage – the aims and research have been completed as stated in our protocol (CRD42023487512). The review was commissioned with a specific purpose: to inform the development of the UK NHS patient safety strategy, particularly the consideration of whether an SMS approach would be appropriate in this policy, organisational and service delivery context. Accordingly, we focused on countries with healthcare systems comparable to the UK NHS. However, given the international comparison in the review, we believe the findings have relevance to an audience beyond the UK. For example, whilst all the included countries have a national healthcare system, they differ in other respects (e.g. we consider the implications of the size of the country in the Discussion)	4 20
 • Unclear study focus The study's aim and focus are unclear. The safety management system is not clearly defined, making it difficult to understand exactly what was investigated and the overall value of the manuscript. The evidence gap is also not clearly articulated; while the authors describe SMS in other industries, they do not specify the evidence base regarding other countries or the current status of healthcare in any country. The rationale for why healthcare	We are unsure in what way the study's aim and focus were felt to be unclear - we already describe and define the main components of a safety management system in the Introduction and in Table 1, based on authoritative sources from the main industries that use them (International Nuclear Safety Advisory Group, 1999; International Civil Aviation Organisation, 2018; OGP, IPIECA, 2014). Moreover, it is this definition that we used to design our searches and inform our data extraction. It is not possible to provide further detail as SMS are specific to the industry and organisation in which they are implemented - we have added a further statement to the Introduction to explain this. We have also added information to the Introduction giving further context on patient safety globally – that one in ten patients receiving hospital care in high income countries are adversely affected, and that the WHO has developed a global action plan to improve safety. This action plan indicates that there are opportunities for healthcare to learn from other high-risk industries where safety is critical, such as aviation. Our commissioning brief, along with our	3, 7-8 4 3

should implement SMS from other industries is unclear, and given that industries often influence each other, such implementation is not surprising. The Introduction should clearly define what is already known and what is unknown.	scoping searches, indicated that the SMS approach used in high-risk industries has influenced the development of patient safety strategies in several countries, including the Netherlands, Ireland, and Australia. However, it was not known if, where SMS have been implemented in healthcare, whether they are effective, and if they are, how to implement them, given international variation in the organisation of healthcare. We do not describe “the current status of healthcare [presumably in relation to patient safety] in any country” because this is the purpose of this review. References International Nuclear Safety Advisory Group. Management of operational safety in nuclear power plants (INSAG-13). Vienna, Austria: International Nuclear Safety Advisory Group; 1999. International Civil Aviation Organisation (ICAOSMM). Safety management manual (Doc 9859, 4th Edition). Montreal, Canada: International Civil Aviation Organisation; 2018. OGP, IPIECA. Operating Management System Framework for controlling risk and delivering high performance in the oil and gas industry, OGP Report No 510. London, UK: OGP & IPIEC; 2014. World Health Organisation (2021) Global patient safety action plan 2021-2030 - Towards eliminating avoidable harm in health care. URL: https://iris.who.int/server/api/core/bitstreams/a28c34c0-089c-4f5d-a0b1-5d9c35a3cd67/content (accessed 15 December 2025).	
 • The authors state, “How are the components of an SMS reflected in the healthcare policy documents?” This is not a suitable question for a systematic review, as it focuses only on nationwide healthcare policy documents. 	These research questions were co-developed and agreed upon with our policy customer—the NHS Patient Safety Team—as well as our Patient and Public Involvement and Engagement group, and are detailed in our protocol. SMS are detailed in policy and strategy documents which is why this aim was included – best practice guidelines for conducting systematic reviews (e.g. CRD) recognise that not all relevant information may be found in the academic literature and previous reviews of policy documents have been conducted (e.g. Just, Tai and Palmier-Claus 2023). Aims 2 and 3 relate to primary research on SMSs. References Centre for Reviews and Dissemination (CRD) (2008) CRD’s guidance for undertaking reviews in health care. URL: https://www.york.ac.uk/media/crd/Systematic_Reviews.pdf (accessed 15 December 2025).	

 • The question, “What does evidence from beyond the UK tell us about the effectiveness, implementation or experience of SMSs within healthcare?” appears targeted at a UK readership. If this is the primary audience, a local journal may be more appropriate. BMJ Open is an international high-impact journal. 	Just, D., Tai, S., & Palmier-Claus, J. (2023). A systematic review of policy and clinical guidelines on positive risk management. Journal of Mental Health, 32(1), 329–340. https://doi.org/10.1080/09638237.2021.1922643 As stated above, our systematic review was commissioned by the NHS Patient Safety Team, and its scope was co-developed with this policy customer and our PPIE group. Its focus, however, is international, comparing the application of SMS in high-income, English-speaking, non-UK countries with healthcare systems similar to that of the UK. We believe, therefore, that its findings are likely to be of interest to a wider audience concerned with patient safety, not just a UK readership. We have added a further statement explaining how the findings can be relevant to a wide readership, as all healthcare systems are working to improve patient safety, and most have some components of an SMS, to the end of the Introduction.	4
Methods  • No inclusion or exclusion criteria, and no search query, are provided in the manuscript. Even if these are described in the supplementary materials, the main criteria should be included in the manuscript. 	We would like to refer the reviewer to sections 2.1 Inclusion and exclusion criteria, and section 2.2 Search strategy and study selection, which can be found in the Methods section in the manuscript. These give information on the inclusion criteria and search strategy, with further detail provided in Supplementary Material 2 and 3.	5
 • Please justify the decision to focus on “rich evidence” and to exclude “thin evidence.” 	We focused on rich evidence as we were investigating the implementation of patient safety nationally, so needed detailed, explanatory accounts of implementation (e.g. showing dynamic relations between changes policies, processes and people/teams). Thin evidence provided more basic descriptive accounts - typically related to evaluations within an organisation, or small group of organisations, and e.g. the accuracy of a patient safety tool rather than its effectiveness - so did not provide the necessary conceptual richness.	9

 • The exclusion of the UK seems inappropriate for a systematic review, as the UK has a national healthcare system and is a high-income country. Other examples of high-income countries with a national healthcare system include Spain (SNS: Sistema Nacional de Salud), Hong Kong (Special Administrative Region of China), and Japan. The rationale for restricting the analysis to only certain countries should be fully explained, and the validity of applying an income-based restriction should also be justified. 	The focus of the review was the application of SMS in healthcare, this meant the UK was not included as we know (and were told by policy-makers) the NHS does not currently use an SMS approach – we have added this information to the Objectives section. The review was restricted to certain countries as (i) they were comparable to the UK, (ii) there was evidence that they were likely to have an SMS from scoping searches, (iii) they were English-speaking countries, or countries where we knew there was definitely an SMS (the Netherlands), as much of the information needed for the review was in policy documents or grey literature, and we had limited ability to translate documents.	4 5
 • The analytical framework for the review is not described in the method. 	We have added a table describing the framework of an SMS that we used for the review, developed from literature on the use of SMS in aviation, oil and gas and nuclear energy.	7-8
Results  • Please summarise the results according to the framework: “Leadership commitment and safety policy,” “Safety risk management,” “Safety assurance,” and “Safety promotion and culture.” If any other results were extracted from the review, please describe them explicitly as well. 	We have described each country in the Results to provide context, so the reader can understand their patient safety approach before they are compared – the justification for the use of case studies, followed by a comparison, is described in section 2.3. Section 3.6, which compares the approaches, is organised according to the framework – we have reorganised and added subheadings to indicate this more clearly, so thank you for your comment.	6 13-19
 • The authors state, “The Netherlands was the only country to explicitly use an SMS approach.” However, 	The patient safety approaches in all countries contained elements that matched with the main components of an SMS (e.g. patient safety policies, risk management). However, in an SMS, these elements are considered as a whole; Australia, Canada, Ireland and New Zealand did not	6

the evidence for this statement is not clear; for example, Australia has the NSQHS. Why is this not considered an SMS approach?	consider their approach to be an SMS, i.e. they were not linking these components together, systematically. We have added to section 2.3 to explain our classification and analysis of the patient safety approaches of these countries.	
 In Table 4, the meaning of the rows is unclear. For example, “Safety assurance” includes two rows, and “Incident reporting and investigation” has three. It would be helpful to include subsection labels for each row so that it is clear what is being described. 	We have added labels to Table 5 (previously Table 4), to explain the subcategories, as well as providing the initial analysis framework in Table 1. We have also added a statement to section 2.3 explaining that the initial framework was revised to allow comparison of the included evidence.	16-18 7-8 6
 The results of the analytical framework—examining the components of an SMS by reviewing policy documents and review articles from high-risk industries—are not clearly presented. 	We hope the revisions made to section 3.6 and Table 5, and the inclusion of Table 1, as described above, improve clarity.	
Discussion  The first paragraph of the Discussion does not appear to be based on the results. 	This paragraph summarises key points from the Results section regarding the patient safety approaches in the included countries.	
 The second paragraph primarily restates the findings rather than interpreting them. The Discussion would be strengthened by elaborating on the implications of these findings—for example, identifying which components of an SMS approach may be most effective in healthcare, and 	This paragraph, along with the preceding paragraph, is intended to summarise the findings, with interpretation provided in the rest of the discussion (which is a typical expectation for structured scientific paper Discussion sections e.g. see: https://www.bmj.com/content/318/7193/1224. We have added to this paragraph to note that the evaluation of SMS in the Netherlands did not look at specific components of an SMS, and as none of the other included countries explicitly used an SMS approach, we are unable to identify which components of an SMS might be most effective.	20

clarifying both the observed impacts of SMS and the anticipated impacts that such an approach aims to achieve.		
 • The third paragraph introduces a new perspective in which healthcare and high-risk industries are viewed as sharing the same theoretical field—safety science—and able to learn from each other. However, no evidence or examples are provided to substantiate this claim. Furthermore, if the relationship is truly reciprocal, this would imply that there are findings or practices from healthcare that could be transferred to high-risk industries, but such examples are not presented. Including concrete evidence or illustrative cases would strengthen the argument. 	We have added a reference to the WHO Global patient safety action plan 2021-2030, which describes parallels between healthcare and high-risk industries and identifies opportunities for learning. The focus of our systematic review was the application of the SMS approach in healthcare, so it was beyond our scope to examine whether high-risk industries have learned from healthcare. Such information, if of interest to the reviewer, can be found in various publications, such as Dekker (2019). Where relevant—for example, in the analysis of the Netherlands’ Patient Safety Programme—we have referenced sources describing the influence of high-risk industries (specifically oil and gas) on the adoption of SMS. As stated in our rationale and aims, the review focused on identifying and describing instances where an SMS approach informed the development of a national patient safety programme directly, as in the Netherlands, or indirectly, through the adoption of specific SMS elements, as in the other four countries. References Dekker, S. (2019). Foundations of Safety Science: A Century of Understanding Accidents and Disasters. Routledge. https://doi.org/10.4324/9781351059794 World Health Organisation (2021) Global patient safety action plan 2021-2030 - Towards eliminating avoidable harm in health care. URL: https://iris.who.int/server/api/core/bitstreams/a28c34c0-089c-4f5d-a0b1-5d9c35a3cd67/content (accessed 15 December 2025).	20
 • The fourth paragraph does not clearly articulate interpretations derived from the study’s findings. This section should explicitly base the discussion on the results of this review and develop arguments that directly emerge from those findings. 	We are unsure what is meant by this statement, as paragraph 4 discusses various concepts and theories used by the included countries, as described in the Results (e.g. the Measurement and Monitoring of Safety Framework in Canada).	

 • In the Discussion, it is important to go beyond reiterating general principles and instead interpret the findings in light of the study’s specific results. The most compelling discussion sections explain how the evidence obtained in the present review confirms, contradicts, or extends existing knowledge, and what implications arise from these findings for policy, practice, or future research. 	We did not find this comment clear – we are unsure what is meant by "interpret the findings in light of the study’s specific results". However, we believe that we have interpreted the evidence gathered and presented within this review, with the most important points being that healthcare systems do not necessarily need an SMS, and that the field of safety science and concepts such as Safety-II, with its focus on understanding success, are increasingly recognised as important in improving patient safety.	
Conclusions  • Overinterpretation of SMS impact SMS-based activities are implemented at the level of individual facilities; therefore, their outcomes should be evaluated at the facility level rather than the national level. Concluding that the SMS approach had an impact only in the Netherlands appears to overinterpret the evidence.	We do not believe this is overinterpreting the evidence, as only the Netherlands implemented an SMS approach. Patient safety in the other countries may have been improved through their approach, but as they did not use SMSs, we have not evaluated them within this review.	
 • Causal inference regarding process safety from other industries The statement that process safety concepts from other industries influenced the approaches in other countries implies a causal relationship not supported by the evidence. A more appropriate	As explained earlier, where relevant, we have provided evidence that the SMS approach, as adopted in high-risk industries, has influenced—directly or indirectly and to varying degrees—the development of national patient safety programmes. We are unclear what is meant by “causal inference” in this context, but we do not claim anywhere that this was the only, or even the main, influence, as it is clear from the field of evidence-based policy that policy developments are products of complex socio-historical processes.	

description might be that the frameworks or ways of thinking from process safety in other industries can be adapted to healthcare, or that similar concepts have been applied in healthcare.		
• Shift toward contextual adaptation and cross-sector learning The assertion that there has been a shift toward recognising the need for contextual adaptation and collaborative learning across industries and sectors does not appear to be directly derived from the results of this study.	It is standard practice for findings from a systematic review to be interpreted within a broader policy and social context. As the reviewer noted earlier, this goes beyond narrowly reporting results and requires drawing on the wider literature and policy trends, even when these were not the primary focus of the review.	
Technical terminology Several key concepts are used without clear definitions—e.g., SMS, concepts from high-risk industries, safety science, and patient safety approach. In this manuscript, “systems approach” appears to refer to the overall conceptual framework of system-oriented safety management, whereas “SMS” refers to the tangible system implemented in practice (e.g., manuals, procedures, reporting systems). However, in the	We define the four key components of an SMS in the Introduction, and have added Table 1, giving more detail on these components. We cannot provide further definition, as SMS differ depending on context (a point on this has also been added in the Introduction). We also have a Glossary which defines term such as safety science, examples of concepts from high-risk industries, and patient safety. We have provided clarification in the manuscript – using a systems approach refers to systems thinking (organisation and process-level factors), with an SMS being one way of operationalising this. More recent safety science approaches, such as Safety II, put equal emphasis on the individual as an agent of change and the broader system in which individuals and teams operate. This is aligned with the broader philosophical understanding of an open system as ‘more than the sum of its parts’, having dynamic relationships and evolving and learning by being in constant exchange with their environment. Reference Hinings, C. R., & Greenwood, R. (2017). The Opening Up of Organization Theory: Open Systems, Contingency Theory, and Organizational Design. In The Oxford Handbook of	3-4 7-8 21-22 3

context of patient safety, “systems approach” is commonly understood in contrast to the “person approach”— meaning a focus on organisational and process-level factors rather than individual blame, aiming to improve the systems and environments that prevent errors. To avoid confusion, I suggest clarifying the intended meaning of “systems approach” in this manuscript.	Management (C. J. Cooper & S. L. Clegg, Eds.), pp. 127–144. Oxford University Press. https://doi.org/10.1093/oxfordhb/9780198708612.013.7	
Questioning the Classification as a Systematic Review While the manuscript is described as a systematic review, the inclusion of only five preselected countries and the use of a predetermined analytical framework raises concerns about whether it fully meets the standards of a systematic review as per PRISMA or Cochrane guidelines. Systematic reviews typically involve comprehensive and transparent searches without geographical restrictions unless these are clearly justified. In this case, the rationale for limiting the scope to these specific countries should be fully explained, and the potential impact of this restriction on the	We consider the manuscript to meet the criteria for being a systematic review defined by Krnic Martinic et al. (2019):  1. Include a clear and pre-specified research question, 2. have used a search strategy that is sufficiently clear and detailed to be reproducible, 3. have pre-specified inclusion/exclusion criteria and screening methods, 4. have conducted quality assessment of included studies, and 5. report a clearly described method of data analysis. Our focus on five countries is justified above and in the manuscript, and the analysis framework is based on the components of an SMS as this was the concept being studied. We have added further information to the Methods section detailing how the framework was used in the analysis and revised based on the included studies. Whilst we did not quality appraise evidence using a formal tool, this was because the majority of included publications were not empirical studies – we did consider the methodological quality of included evaluations when reporting their results. We had acknowledged that the focus on high-income, English-speaking countries is a limitation of the review in the Discussion, and have now added further detail to this revision of the manuscript (that lower income and non-English speaking countries may be implementing SMS). Reference Krnic Martinic M, Pieper D, Glatt A, Puljak L. Definition of a systematic review used in overviews of systematic reviews, meta-epidemiological studies and textbooks. BMC Medical Research Methodology. 2019;19:203.	3-4 6, 7-8 19 21

generalisability of the findings should be discussed. The recommendation is to change this to a scoping review, or to expand the inclusion criteria for countries, or to limit the scope based on health insurance system type.		
Reviewer 2		
The manuscript summarizes the application of SMS in five countries and is a useful for understanding the situation in developed countries. There are no major comment with the content. As this is a study commissioned by NHS England, it is understandable that it was written with the UK in mind. It would be helpful to include specific examples of where the findings could be applied. P6L40, section 2.3 should be 2.4.	Thank you for reviewing the manuscript and noting the error. section 2.3, this has been corrected. We have added a further statement to the Introduction explaining that we believe the findings will be of interest to a broad readership—beyond the UK and the five countries included in the review—as they offer insights into the usefulness of the SMS approach as a framework or source of inspiration for developing national patient safety programmes. The cross-country comparison highlights the specific considerations required when adapting such a relatively rigid approach to different national contexts and the challenges likely to arise. It also sheds light on the difficulties associated with evaluating and further developing programmes that are continuously evolving.	9 4
Reviewer 3	We have not edited the manuscript based on this review as the comments refer to a paper on expert leadership rather than our research on safety management systems.	

VERSION 2 - REVIEW

Reviewer **1**

Name **Koike, Daisuke**

Affiliation
Public Health

Fujita Health University School of Medicine, Department of

Date

25-Jan-2026

COI

I carefully reviewed the revised manuscript and the authors' responses. While some minor clarifications have been made, the major methodological and interpretive concerns raised in the initial review remain largely unaddressed. In particular, key issues regarding study aim, design, and country selection continue to be justified primarily by commissioning context rather than scientific rationale, and fundamental principles of systematic review methodology are not met. The synthesis remains descriptive and case-study-like, and major concerns regarding the Discussion and Conclusion remain unresolved. I therefore recommend rejection.

General comment

1. Need for a more focused aim and alignment of results

I previously raised concerns about the lack of a clearly justified and internationally relevant study scope. Protocol adherence cannot compensate for a narrowly and inconsistently justified scope, particularly in a study presented as an academic systematic review rather than a policy briefing. The rationale for selecting only five countries as inclusion criteria was not sufficiently provided.

2. Unclear study focus

The authors' response reiterates that SMS are defined based on other industries; however, this does not resolve the core concern. However, it is unclear whether this definition is sufficiently specific to identify and extract SMSs in the healthcare context. Given the conceptual overlap with other healthcare frameworks (e.g., quality management systems or patient safety programs), a more detailed and operational definition appears necessary. According to the authors, an explicitly implemented SMS in healthcare was identified only in the Netherlands. This creates an internal inconsistency: if the review targets only explicitly implemented SMSs, the Netherlands would be the sole eligible country. Conversely, if the study aim is to explore SMS-related approaches in healthcare more broadly, including systems that align with SMS principles even if not explicitly labelled as such, this broader scope needs to be clearly defined and justified.

3. Questioning the Classification as a Systematic Review

Despite the authors' assertion, the search strategy is not described with sufficient detail to be considered reproducible. In particular, the absence of clearly reported search terms is a critical limitation. Furthermore, a substantial proportion of the included evidence appears to have been identified through internet searches; however, the methods used for web-based searching are not described.

In addition, the approach used to restrict the search to specific countries is not clearly specified. The criterion of being “comparable to the UK” is difficult to operationalise in a reproducible manner. As noted previously, explicit exclusion criteria are also not clearly defined.

Finally, the data analysis is described as a case study approach. This is not a standard analytical method for systematic reviews, and its methodological justification in this context is not sufficiently explained. Taken together, these issues raise substantial concerns about whether the manuscript meets the methodological expectations of a systematic review.

Specific comment

4. I previously raised concerns that framing the mapping of SMS components within national healthcare policy documents is not a suitable research question for a systematic review, as stated in the manuscript: “How are the components of an SMS reflected in the healthcare policy documents?”. The comment has not been addressed. Framing the mapping of SMS components within national policy documents as a core research question remains methodologically inconsistent with a systematic review.

5. I previously raised concerns that the research question “What does evidence from beyond the UK tell us about the effectiveness, implementation or experience of SMSs within healthcare?” is framed from a UK-centric perspective and is not appropriate for an international journal. This comment has not been addressed, as the question remains UK-centric and reads as a reflective statement rather than a clearly defined research objective suitable for the Introduction.

6. Inclusion and exclusion criteria

While the authors describe the types of evidence included, explicit inclusion and exclusion criteria are not clearly specified, particularly regarding what was excluded and on what basis. This limits assessment of the review’s reproducibility and methodological rigour. In addition, for a study presented as a systematic review, the search strategy—including key search terms and search strings—should be transparently reported, ideally in the Supplementary Materials, to allow evaluation of reproducibility and potential selection bias.

7. Rich / thin evidence definition

As previously raised, the criteria used to distinguish between “rich” and “thin” evidence appear subjective, and their application is difficult to reproduce. It also remains unclear whether this distinction was defined a priori in the protocol or applied post hoc during study selection.

8. The exclusion of the UK

As previously raised, the exclusion of the UK and the restriction of the review to a small number of selected countries require clear methodological justification in a systematic review. Restricting the review to five countries based on UK comparability, income level, and

scoping results is not consistent with the study title or stated objectives. This misalignment between the declared scope and the eligibility criteria undermines the systematic nature of the review.

9. Discussion

I raised multiple comments on the Discussion in the initial review, particularly regarding the lack of interpretation grounded in the review’s specific finding. These concerns have not been substantively addressed in the revised manuscript. The Discussion continues to restate findings or present general theoretical perspectives without clearly articulating what new insights emerge from this review. As a result, the Discussion does not fulfil its role of interpreting the findings in relation to existing knowledge and their implications.

10. Conclusion

I raised several concerns regarding the Conclusion, which have not been adequately addressed. My concern is not whether only the Netherlands implemented an SMS, but rather the level at which impact is inferred. As SMSs are implemented primarily at the organisational or facility level, attributing changes in patient safety outcomes at the national level to an SMS approach risks overinterpretation, particularly given the heterogeneity in implementation and evaluation. In addition, wording such as “had some influence” and statements that the findings “reflect a shift” toward contextual adaptation and cross-sector learning may be read as implying causal or systematic change, whereas the evidence more consistently supports conceptual alignment or adaptation of ideas. Clarifying these distinctions in the Conclusion would help avoid overstating what can be inferred from the review’s findings.

11. Additional comment

The authors list “What research or other relevant evidence is available regarding the effectiveness, implementation or experience of SMSs within healthcare?” as one of the study objectives. However, the Results section does not provide a synthesis of evidence addressing effectiveness, implementation, or experience. As currently presented, it is unclear how this objective was examined or answered within the review.

VERSION 2 - AUTHOR RESPONSE

Comment	Response
Reviewer 1	
Comments to the Author: I carefully reviewed the revised manuscript and the authors’ responses. While some minor	Whilst our approach to this systematic review resulted from the research commissioning context, all decisions made regarding the conduct of the review had a scientific rationale and are consistent with the highest standards of systematic review

clarifications have been made, the major methodological and interpretive concerns raised in the initial review remain largely unaddressed. In particular, key issues regarding study aim, design, and country selection continue to be justified primarily by commissioning context rather than scientific rationale, and fundamental principles of systematic review methodology are not met. The synthesis remains descriptive and case-study-like, and major concerns regarding the Discussion and Conclusion remain unresolved. I therefore recommend rejection.	methodology – below, and in the revised manuscript, we provide further detail explaining our approach.
General comment 1. Need for a more focused aim and alignment of results I previously raised concerns about the lack of a clearly justified and internationally relevant study scope. Protocol adherence cannot compensate for a narrowly and inconsistently justified scope, particularly in a study presented as an academic systematic review rather than a policy briefing. The rationale for selecting only five countries as inclusion criteria was not sufficiently provided.	Our main criterion for inclusion was that the country was implementing safety management systems (SMS) in their healthcare system. As we were comparing the implementation of SMS, our other criteria related to enabling comparison by looking at countries that had similarities in terms of being high income and the organisation and financing of their healthcare systems, i.e. universal coverage (via general taxation or social insurance) (see references below). This focus meant our scope was not initially narrow, but as our scoping searches suggested that only Ireland, Australia, New Zealand, Canada and the Netherlands used an SMS approach (or component principles), we specified that we would restrict the review to evidence from these countries (p5). Whilst screening, we found that only the Netherlands had used an SMS approach; the decision to continue including the other countries is explained further below and in the manuscript (p6). We have also added further detail explaining the rationale for restricting the review to these countries (p5). We acknowledge there may be non-English speaking countries that were not included, but we did not have the resource to translate documents and acknowledge this as a limitation of the review (p22).

	References European Observatory on Health Systems and Policies. Netherlands. 2016. URL: https://eurohealthobservatory.who.int/countries/netherlands/ (accessed 26 February 2026). The Commonwealth Fund. Country profiles - International health care system profiles. 2020. URL: https://www.commonwealthfund.org/international-health-policy-center/countries (accessed 26 February 2026).
2. Unclear study focus The authors' response reiterates that SMS are defined based on other industries; however, this does not resolve the core concern. However, it is unclear whether this definition is sufficiently specific to identify and extract SMSs in the healthcare context. Given the conceptual overlap with other healthcare frameworks (e.g., quality management systems or patient safety programs), a more detailed and operational definition appears necessary. According to the authors, an explicitly implemented SMS in healthcare was identified only in the Netherlands. This creates an internal inconsistency: if the review targets only explicitly implemented SMSs, the Netherlands would be the sole eligible country. Conversely, if the study aim is to explore SMS-related approaches in healthcare more broadly, including systems that align with SMS principles even if not explicitly labelled as	In the previous revision, we added Table 1 to the manuscript, this defines what we considered to be an SMS based on our understanding of the literature, providing more detail on subcategories contained within each of the four main components. These subcategories (e.g. policy documents, risk assessment, safety investigations, safety culture, etc.) are all found within healthcare systems and formed an operational definition - as stated in section 2.1 Inclusion and exclusion criteria, we considered evidence relating to any of these components eligible for inclusion. We also describe (p6) how it was only when screening we found that countries other than the Netherlands did not use an SMS approach. We agree that this would mean only the Netherlands was eligible for inclusion based on our initial criteria. However, stakeholder consultation indicated that a comparison with the safety approaches of other countries would be useful in understanding the potential impact of an SMS approach. We have edited this section to more clearly indicate that this was a change to the protocol (p6).

such, this broader scope needs to be clearly defined and justified.	
3. Questioning the Classification as a Systematic Review Despite the authors' assertion, the search strategy is not described with sufficient detail to be considered reproducible. In particular, the absence of clearly reported search terms is a critical limitation. Furthermore, a substantial proportion of the included evidence appears to have been identified through internet searches; however, the methods used for web-based searching are not described. In addition, the approach used to restrict the search to specific countries is not clearly specified. The criterion of being "comparable to the UK" is difficult to operationalise in a reproducible manner. As noted previously, explicit exclusion criteria are also not clearly defined. Finally, the data analysis is described as a case study approach. This is not a standard analytical method for systematic reviews, and its methodological justification in this context is not sufficiently explained. Taken together, these	Please see the Supplementary Material fully describing our searches (as previously supplied), which provides the exact search terms used for the database searches. They also provide detail of the websites searched, including the location of search (URL), search terms, dates searched and number of hits. [NB. Our very experienced information specialist (co-author Bethel) who developed, ran and documented our searches is well aware of the importance of and expected conventions for fully reporting all or any types of searches conducted for publishing systematic reviews. Some of her international contributions to these methods are listed below] As described above and in the manuscript (p5), we have provided more detail on what is meant by 'comparable to the UK'. We also explain the case study approach below; this was intended to provide describe the patient safety approach in each country, providing contextual information and allowing comparison. Further detail on the inclusion and exclusion criteria is in the Supplementary Material (which was included in the previously supplied materials). While we agree that a case study approach is perhaps not a standard or commonly explicit method within systematic reviews, it is often the case that studies included in a systematic review come from a limited range of countries AND where the clinical and/or policy context is so relevant to the review question it makes more sense to synthesise the evidence from each country separately – at least initially. Furthermore, our review is not a focused review of the effectiveness of a simple intervention, it is answering a complex policy question, and can therefore be considered a complex review (Mahtani et al. 2018). Methods papers on complex interventions and systematic reviews describe the importance of contextual information and that non-standard methods may need to be used to synthesise the evidence (Mahtani et al. 2018; Petticrew et al. 2013). Policy and other documents (i.e. non-research) are important for understanding the content, evolution and complexity of health policy (Daglish et al. 2020; Kayesa & Shung-King, 2021), with the use of case studies recognised as an appropriate approach for document analysis (Daglish et al. 2020). We used a case study approach in order to provide a full description and contextual information on the patient safety approach in each country, as needed to understand how they relate to SMS, and to compare them. We have added to our explanation of the case study approach to include this justification (p6).

issues raise substantial concerns about whether the manuscript meets the methodological expectations of a systematic review.	References DalGLISH SL, Khalid H, McMahon SA. Document analysis in health policy research: the READ approach, Health Policy and Planning, Volume 35, Issue 10, December 2020, Pages 1424–1431, https://doi.org/10.1093/heapol/czaa064 Mahtani KR, Jefferson T, Heneghan C, et al. What is a ‘complex systematic review’? Criteria, definition, and examples. BMJ Evidence-Based Medicine 2018;23:127-130. Petticrew M, Anderson L, Elder R, Grimshaw J, Hopkins D, Hahn R, et al. Complex interventions and their implications for systematic reviews: a pragmatic approach. Journal of Clinical Epidemiology 2013;66:1209-14. https://doi.org/https://doi.org/10.1016/j.jclinepi.2013.06.004 Our information specialist’s international contributions to the conduct and reporting of literature search methods: Bethel, A. C., Rogers, M., & Abbott, R. (2021). Use of a search summary table to improve systematic review search methods, results, and efficiency. Journal of the Medical Library Association : JMLA, 109(1), 97–106. https://doi.org/10.5195/jmla.2021.809 Rogers, M., Sutton, A., Campbell, F., Whear, R., Bethel, A., & Coon, J. T. (2024). Streamlining search methods to update evidence and gap maps: A case study using intergenerational interventions. Campbell systematic reviews, 20(1), e1380. https://doi.org/10.1002/cl2.1380 Rogers, M., Bethel, A., & Boddy, K. (2017). Development and testing of a medline search filter for identifying patient and public involvement in health research. Health information and libraries journal, 34(2), 125–133. https://doi.org/10.1111/hir.12157
Specific comment 4. I previously raised concerns that framing the mapping of SMS components within national healthcare policy documents is not a suitable research question for a systematic review, as stated in the manuscript: “How are the components of an SMS reflected in the healthcare policy documents?”. The	We responded to this comment previously - SMS are detailed in policy and strategy documents, not in journal articles, which is why this aim was included and the inclusion criteria are not restricted to research studies. Best practice guidelines for conducting systematic reviews (e.g. Centre for Reviews and Dissemination guidance, the Cochrane Handbook) recognise that not all relevant information for particular questions may be found in the academic literature. Policy documents are important for understanding the content, evolution and complexity of health policy (DalGLISH et al. 2020; Kayesa & Shung-King, 2021), and previous reviews of policy documents have been conducted (e.g. Just, Tai and Palmier-Claus 2023). To address this comment further, we have described above how this is a complex review - Petticrew et al. (2013) state that “Not all

comment has not been addressed. Framing the mapping of SMS components within national policy documents as a core research question remains methodologically inconsistent with a systematic review.	useful sources of information about complexity are scientific studies”, and that policy and other documents provide important contextual information. Whilst the inclusion of policy and other documents is unconventional – and certainly would not be suitable for reviews of the effectiveness of specific interventions - for questions relating to health policy, and understanding service delivery at a national level, we believe this is an appropriate approach. We have added that policy documents were included as they were essential to understanding the patient safety approach in each country to the inclusion criteria (p5). References Centre for Reviews and Dissemination (CRD) (2008) CRD’s guidance for undertaking reviews in health care. URL: https://www.york.ac.uk/media/crd/Systematic_Reviews.pdf (accessed 15 December 2025). DalGLISH SL, Khalid H, McMahon SA. Document analysis in health policy research: the READ approach, Health Policy and Planning, Volume 35, Issue 10, December 2020, Pages 1424–1431, https://doi.org/10.1093/heapol/czaa064 Higgins JPT, Thomas J, Chandler J, Cumpston M, Li T, Page MJ, et al, editor(s). Cochrane Handbook for Systematic Reviews of Interventions version 6.5 (updated August 2024). Cochrane, 2024. Available from www.cochrane.org/handbook. Just, D., Tai, S., & Palmier-Claus, J. (2023). A systematic review of policy and clinical guidelines on positive risk management. Journal of Mental Health, 32(1), 329–340. https://doi.org/10.1080/09638237.2021.1922643 Kayesa NK, Shung-King M. (2021) The role of document analysis in health policy analysis studies in low and middle-income countries: Lessons for HPA researchers from a qualitative systematic review. Health Policy; 2: 100024. Petticrew M, Anderson L, Elder R, Grimshaw J, Hopkins D, Hahn R, et al. Complex interventions and their implications for systematic reviews: a pragmatic approach. Journal of Clinical Epidemiology 2013;66:1209-14. https://doi.org/https://doi.org/10.1016/j.jclinepi.2013.06.004
5. I previously raised concerns that the research question “What does evidence from beyond the UK tell us about the effectiveness, implementation or experience of SMSs	As explained previously, this research question was predefined in our protocol. Whilst the phrasing is UK-centric, it states clearly that an objective of the review will be to evaluate evidence on the effectiveness, implementation or experience of SMSs within healthcare.

within healthcare?” is framed from a UK-centric perspective and is not appropriate for an international journal. This comment has not been addressed, as the question remains UK-centric and reads as a reflective statement rather than a clearly defined research objective suitable for the Introduction.	
6. Inclusion and exclusion criteria While the authors describe the types of evidence included, explicit inclusion and exclusion criteria are not clearly specified, particularly regarding what was excluded and on what basis. This limits assessment of the review’s reproducibility and methodological rigour. In addition, for a study presented as a systematic review, the search strategy—including key search terms and search strings—should be transparently reported, ideally in the Supplementary Materials, to allow evaluation of reproducibility and potential selection bias.	Please see the Supplementary Material, which provide the exact search terms used for the database searches and website searches. They also contain the inclusion and exclusion criteria. The scope of this review – national patient safety approaches – was broad, so there were few exclusion criteria. We have also added a statement to the manuscript to on exclusion criteria, stating that studies were excluded where their focus was not linked to a component of an SMS, or was not patient safety (e.g. occupational health and safety) (p5).
7. Rich / thin evidence definition As previously raised, the criteria used to distinguish between “rich” and “thin” evidence appear subjective, and their application is difficult to	We have added to the manuscript to explain that this was post-hoc change to the review (p6), and that this method of classifying studies to enable analysis is an accepted technique for systematic reviews (Ames et al. 2019). The approach we used to classify studies was by data richness and closeness of the study data to the review objective, as in Ames et al. (2019), and we have also added this information to the manuscript (p6). We provide the full list of included studies, separated into those considered

reproduce. It also remains unclear whether this distinction was defined a priori in the protocol or applied post hoc during study selection.	rich and those considered thin, which, along with the provided definition of rich and thin studies, should provide sufficient detail for reproducibility. References Ames HMR, Glenton C, Lewin S, Tamrat T, Akama E, Leon N. Clients' perceptions and experiences of targeted digital communication accessible via mobile devices for reproductive, maternal, newborn, child, and adolescent health: a qualitative evidence synthesis. Cochrane Database of Systematic Reviews 2019; 10.1002/14651858.CD013447:CD013447. https://doi.org/10.1002/14651858.CD013447
8. The exclusion of the UK As previously raised, the exclusion of the UK and the restriction of the review to a small number of selected countries require clear methodological justification in a systematic review. Restricting the review to five countries based on UK comparability, income level, and scoping results is not consistent with the study title or stated objectives. This misalignment between the declared scope and the eligibility criteria undermines the systematic nature of the review.	We have provided further detail on the restriction of the review to five countries above. The exclusion of the UK is justified as the main criteria for inclusion in the review was the use of SMS in the healthcare system – the UK's national approach to patient safety does not involve SMS.
9. Discussion I raised multiple comments on the Discussion in the initial review, particularly regarding the lack of interpretation grounded in the review's specific finding. These concerns have not been substantively addressed in the revised manuscript. The Discussion continues to	We believe that we have interpreted the evidence gathered and presented within this review in the context of the wider literature. We found limited evidence on the effectiveness, implementation or experience of different patient safety approaches (p20), so believe our comments on these points regarding the need for better evaluation are appropriate. The main findings we can draw from the review regard the changes in safety science and concepts used to inform national approaches - that healthcare systems do not necessarily need an SMS, and that the field of safety science and concepts such as Safety-II, with its focus on understanding success, are increasingly recognised as important in improving patient safety.

restate findings or present general theoretical perspectives without clearly articulating what new insights emerge from this review. As a result, the Discussion does not fulfil its role of interpreting the findings in relation to existing knowledge and their implications.	
10. Conclusion I raised several concerns regarding the Conclusion, which have not been adequately addressed. My concern is not whether only the Netherlands implemented an SMS, but rather the level at which impact is inferred. As SMSs are implemented primarily at the organisational or facility level, attributing changes in patient safety outcomes at the national level to an SMS approach risks overinterpretation, particularly given the heterogeneity in implementation and evaluation. In addition, wording such as “had some influence” and statements that the findings “reflect a shift” toward contextual adaptation and cross-sector learning may be read as implying causal or systematic change, whereas the evidence more consistently supports conceptual alignment or adaptation of ideas. Clarifying	The Netherlands required every public hospital in the country to implement an SMS approach (see p11), so impacts at a local level would also be expected to be seen nationally. When describing the evaluation of the patient safety programme in the Netherlands, we acknowledge that there was variation, and have also added a statement to clarify that the evaluation was of the whole patient safety programme (p20). As we say that there ‘some longitudinal evidence of improving patient safety’ from the Netherlands, we do not feel this is overstating the findings of the evaluation. We have made changes to the Conclusion so that is clear we are not implying a causal relationship regarding the safety concepts informing national patient safety approaches (p22).

these distinctions in the Conclusion would help avoid overstating what can be inferred from the review's findings.	
11. Additional comment The authors list "What research or other relevant evidence is available regarding the effectiveness, implementation or experience of SMSs within healthcare?" as one of the study objectives. However, the Results section does not provide a synthesis of evidence addressing effectiveness, implementation, or experience. As currently presented, it is unclear how this objective was examined or answered within the review.	There was very limited evidence relating to this objective, with only three countries conducting national evaluations of their patient safety approach. This evidence is synthesised in the manuscript (p20), and we have added a statement indicating that it answers this objective.